# Towards Foundation Models for Zero-Shot Time Series Anomaly Detection: Leveraging Synthetic Data and Relative Context Discrepancy

**Tian Lan** [* 1]  **Hao Duong Le** [* 1]  **Jinbo Li** [* 1]  **Wenjun He** [2]  **Meng Wang** [2]  **Chenghao Liu** [3]  **Chen Zhang** [1]

## Abstract

Time series anomaly detection(TSAD) is a critical task, but developing models that generalize to unseen data in a zero-shot manner remains challenging. Existing foundation models for TSAD often rely on reconstruction-error scoring at inference time, which can miss subtle anomalies that are well reconstructed and can falsely flag complex but normal patterns in unseen domains. We introduce TimeRCD, a foundation model for TSAD built on Relative Context Discrepancy (RCD), a pre-training paradigm that trains the model to detect anomalies by comparing a query pattern with its surrounding context. This relational formulation, implemented with a standard Transformer architecture, enables the model to infer normality from the input context rather than relying on fixed global normal patterns. We further construct a large-scale synthetic corpus with context-dependent anomaly labels to provide supervised pre-training signals for RCD. Experiments across diverse benchmarks show that TimeRCD outperforms existing general-purpose and anomaly-specific foundation models in most zero-shot TSAD settings, while remaining competitive with dataset-specific full-shot baselines. These results provide empirical evidence that RCD is an effective direction for building robust and generalizable TSAD models.

## 1. Introduction

Time series anomaly detection (TSAD) is a crucial task in domains such as finance (Ahmed et al., 2016), health-

*Equal contribution [1]Department of Industrial Engineering, Tsinghua University, Beijing, China [2]2012 Lab, Huawei Technologies, ShenZhen, China [3]Datadog AI Research, Paris, France. This work was completed prior to joining Datadog. Correspondence to: Chen Zhang <zhangchen01@tsinghua.edu.cn>, Chenghao Liu <twinsken@gmail.com>.

*Proceedings of the 43$^{rd}$ International Conference on Machine Learning*, Seoul, South Korea. PMLR 306, 2026. Copyright 2026 by the author(s).

care (Kaji et al., 2019), industrial monitoring (Lan et al., 2025), and cloud operations (Ren et al., 2019). The accurate detection of rare and unexpected events is vital for ensuring system reliability and safety. Despite recent progress driven by deep learning, most existing approaches are trained in a dataset- and model-specific manner, which restricts their scalability and hampers generalization across diverse domains in a *zero-shot* way.

The success of foundation models in natural language processing and computer vision has motivated efforts to establish similar paradigms for TSAD. Existing approaches can be broadly categorized into two directions: (i) general-purpose time series foundation models designed for multiple tasks such as classification, forecasting, and anomaly detection (Gao et al., 2024; Goswami et al., 2024; Woo et al., 2024; Ekambaram et al., 2025; Xie et al., 2024), and (ii) anomaly-specific foundation models tailored explicitly for TSAD (Shentu et al., 2024). Additional related work is discussed in Appendix A. Despite their differences, both types of models predominantly rely on reconstruction-based objectives trained on real-world data, where anomalies are inferred indirectly from *reconstruction errors*.

While intuitive, reconstruction-error inference can suffer from a fundamental **objective mismatch**: models are optimized to reconstruct normal patterns in a latent space where anomalous structure is assumed to be lost (Wong et al., 2022). This leads to critical limitations, as illustrated in Figure 1(a,b). First, subtle and contextual anomalies are often smoothed out and thus missed, resulting in low reconstruction error and false negatives (Wong et al., 2022; Wu et al.). Second, complex but normal sequences deviate from the "average" patterns learned during training, yielding high reconstruction errors and false alarms (Yahya et al., 2025). While recent works (Yang et al., 2023; Xu et al., 2021) have moved beyond simple reconstruction errors by employing discrepancy-based mechanisms, they fundamentally hinge on a **global normality assumption**. These methods are optimized to learn a fixed distribution of normality predominantly from the training set. However, such paradigms are ill-suited for *zero-shot* settings due to what we term **representation conflict**. Since the definition of normality often undergoes a *concept shift* across domains, a morphological

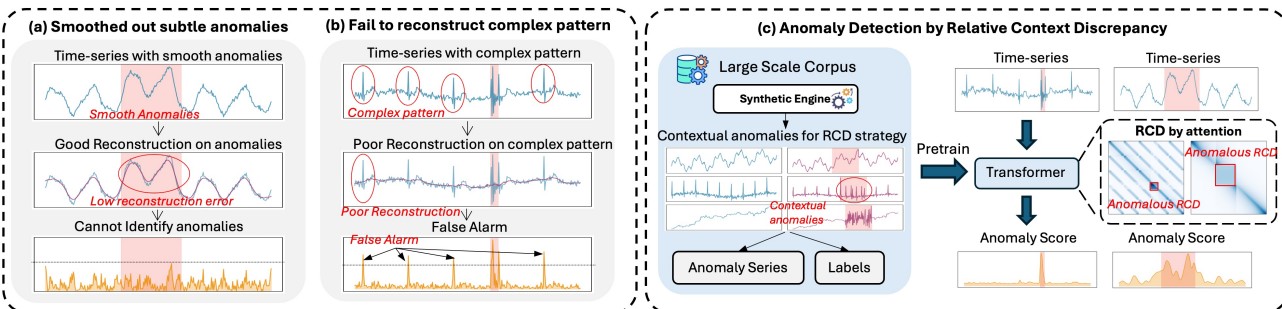

Figure 1. **Limitations of Reconstruction-based TSAD and Our Proposed TimeRCD.** (a) The model accurately reconstructs a smooth anomaly, resulting in a low error score and a missed detection (false negative). (b) The model fails to reconstruct a complex normal pattern unseen in the training dataset, leading to a high error score and a false alarm (false positive). **(c)** TimeRCD adopts RCD with a large-scale context-dependent corpus and a standard encoder-only Transformer.

pattern considered normal in one domain may be inherently anomalous in another domain (Fig. 2). The model faces optimization conflicts and poor generalization when compelled to map identical patterns to contradictory labels within a unified latent space. We emphasize that this paper does not reject reconstruction as a pretext task: our concern is the common inference rule that treats reconstruction error itself as the anomaly score.

This **representation conflict** is difficult to resolve with standard supervision strategies. First, labeled anomalies are inherently **scarce and lack diversity**, providing the model with few examples to learn abnormal behavior. Second, existing methods attempt to mitigate limited supervision via data augmentation (Shentu et al., 2024; Darban et al., 2025; Cai et al., 2024). However, a critical limitation remains: these techniques primarily focus on expanding **morphological diversity** (e.g., via jittering or masking) but **often struggle to** break the semantic coupling between local patterns and their normality labels. They **risk reinforcing** the bias that "specific shapes are intrinsically normal," rather than teaching the model to evaluate patterns relative to their surroundings. Consequently, **paradigms** that rely on **memorizing fixed normal patterns**—even with augmented data—**face intrinsic limitations** in zero-shot scenarios, highlighting the need to decouple the definition of normality from specific morphological features.

To address these limitations, we introduce TimeRCD, a novel foundation model for TSAD built on a new pre-training paradigm, as illustrated in Figure 1(c). Our approach replaces reconstruction-error scores at inference time by explicitly training the model to detect anomalies through the **Relative Context Discrepancy** (RCD) paradigm in zero-shot settings. Masked reconstruction is retained only as an auxiliary training signal for temporal representation learning, while the final anomaly decision is made by a dedicated anomaly head. Instead of memorizing static normal patterns from the training dataset, we train the model to **adaptively identify** normality directly from the unlabeled input sequence itself. The model identifies intrinsic normal patterns

within the immediate context, using them as references to detect anomalies on the fly without relying on pre-defined training distributions.

For our pre-training process, we employ a standard Transformer backbone without any architectural modifications. We tokenize each time series into non-overlapping variate-window patches and project them into Transformer input tokens, which allows the self-attention mechanism to naturally compute inter-token relationships, as shown in Figure 1(c). To teach the model this explicit RCD strategy, we provide it with a rich, supervised signal by first leveraging a synthetic engine that breaks the semantic coupling between local patterns and their normality labels. This engine generates a large-scale, diverse, and context-dependent corpus of time series data that is specifically designed to contain a wide variety of context-dependent anomalies, enabling the model to learn the RCD task from the ground up. Our main contributions are threefold:

- **RCD paradigm for zero-shot TSAD.** We introduce a novel pre-training paradigm that moves beyond reconstruction-error inference by explicitly distinguishing anomalies through the RCD paradigm. This design enables a standard Transformer to capture normality, achieving strong generalization without architectural complexity.

- **A large-scale, context-dependent synthetic corpus for breaking semantic coupling.** To support the RCD pre-training paradigm, we construct a comprehensive synthetic corpus specifically populated with context-dependent deviations. It provides token-level annotations for a diverse spectrum of anomalies, including point, contextual, and collective types with cross-variate propagation, offering the essential supervision required for the model to learn the RCD mechanism.

- **Extensive empirical evaluation** Experiments on diverse corpora demonstrate consistent gains over existing reconstruction-based and general-purpose time-

series foundation models. Ablation studies confirm the contributions of both the synthetic corpus and the RCD framework to zero-shot performance.

## 2. Methodology: TimeRCD Framework

### 2.1. Preliminaries

**Problem Definition** For the zero-shot time series anomaly detection problem, we observe a multivariate $d$-channel time series $\mathbf{X} = (\mathbf{x}_1, \ldots, \mathbf{x}_n)$ with $\mathbf{x}_t \in \mathbb{R}^d$ for each time step $t \in [n] := \{1, 2, \ldots, n\}$. The objective is to produce a binary annotation sequence $\hat{\mathbf{y}} = (\hat{y}_1, \ldots, \hat{y}_n) \in \{0, 1\}^n$ such that $\hat{y}_t = 1$ if and only if time $t$ is anomalous. In the zero-shot setting, the model must detect anomalies on unseen target sequences without any additional target-domain training, distinguishing normal from anomalous behavior.

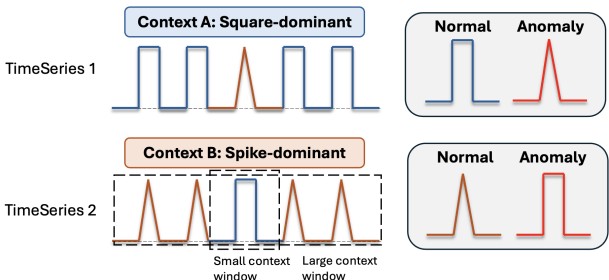

*Figure 2.* Motivation Examples for RCD: Representation conflict in zero-shot TSAD.

**RCD for zero-shot TSAD** Prevailing paradigms typically define anomalies based on deviations from a *global* normality learned from training data. However, this assumption falters in zero-shot settings due to **representation conflict**, as illustrated in Fig. 2. **Consider the role of the "square wave":** in the top sequence, it establishes the dominant normal rhythm, rendering the spike anomalous. Conversely, in the bottom sequence, the roles are reversed—the spike constitutes the normal baseline, while the square wave becomes the anomaly. This elementary example demonstrates that in zero-shot settings, "normality" is not an intrinsic property of a pattern's morphology but a *relative concept* defined purely by its context. To address this, we aim to shift the objective from *remembering* global normality to explicitly *assessing* context-level normality, enabling the model to define and detect anomalies primarily from contextual regularities observed in the target sequence at inference time.

To address this representation conflict, RCD is built on two complementary components: large-context comparative modeling and context-dependent synthetic supervision. Together, they enable the model to infer normality in a context-relative manner at inference time: (1) **Large-context comparative modeling.** Establishing a reliable "normality reference" requires sufficient contextual support

and the ability to compare local temporal patterns within the context. As illustrated in Fig. 2, whether a square wave constitutes normal behavior is only identifiable when long-range repetitions and contextual temporal structure are observable. A large context window allows the model to capture such long-term regularities and periodic patterns, forming an internal baseline against which transient deviations can be evaluated. (2) **Context-Dependent Synthetic Supervision.** To prevent the model from degenerating into memorizing specific morphological features (e.g., falsely learning that "flat lines are always normal"), the training process must explicitly decouple pattern appearance from its *contextual role* (i.e., whether a local pattern is considered normal or anomalous under a given context). We achieve this by constructing synthetic training data in which identical patterns are assigned different contextual roles across varying contexts—appearing as normal in some sequences and anomalous in others. This context-dependent supervision encourages the model to move beyond surface-level pattern recognition and instead learn the underlying principle of contextual comparison.

### 2.2. RCD Strategy and Foundation Model Architecture

**Standard Transformer for large-context comparative modeling.** To implement the first component of the RCD paradigm—capturing long-range contextual dependencies—we employ a standard encoder-only Transformer architecture (Vaswani et al., 2017) without structural modifications, as illustrated in Fig. 3. Input time series are tokenized into non-overlapping patches, enabling the model to process long contexts within a feasible computational budget. Crucially, the Transformer's native self-attention mechanism allows each token to contextualize itself with respect to the entire observation window, thereby enabling implicit comparisons between local temporal patterns and the dominant structure of the sequence. Furthermore, we provide the detailed mathematical formulation of RCD and discuss its theoretical connection to attention mechanisms in Appendix C.

**Variate-Window Tokenization** Since multivariate anomaly detection is a critical task (Zamanzadeh Darban et al., 2024), we build on the design introduced by Moirai (Woo et al., 2024), which flattens multivariate time series so that all variates are represented within a single sequence. This design allows the subsequent Transformer blocks to capture both intra-variate dependencies and inter-variate dependencies. Specifically, given a normalized multivariate time series $\tilde{\mathbf{X}} \in \mathbb{R}^{n \times d}$, we partition it into $\lceil n/W \rceil \times d$ non-overlapping windows, where $W$ denotes the window length. The resulting windows are then flattened and linearly projected into input token embeddings $\mathbf{H}_{inp} \in \mathbb{R}^{\lceil n/W \rceil d \times D_v}$.

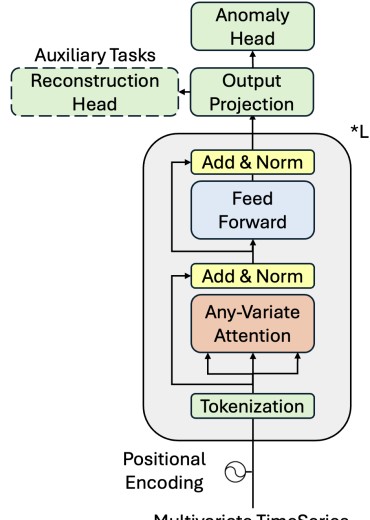

*Figure 3.* TimeRCD Architecture

**Transformer Blocks** We stack $L$ Transformer blocks, each consisting of layer normalization, feed-forward network, and self-attention modules. The architecture imposes no special requirements on the Transformer block itself. Specifically, we leverage any-variate attention (Woo et al., 2024) and formulate the output token embeddings as $\mathbf{H}_{out} \in \mathbb{R}^{\lceil n/W \rceil d \times D_v}$.

**Output Projection and Anomaly/Reconstruction Head** To derive anomaly scores at the original temporal resolution, the output token embeddings $\mathbf{H}_{out}$ from the Transformer blocks are projected back into the observation space. Concretely, we define two heads during training: $\mathbf{X}_{rec} = \text{Reshape}(\mathbf{H}_{out}\mathbf{W}_s\mathbf{W}_{rec}) \in \mathbb{R}^{T \times d}$ and $\mathbf{X}_{ano} = \text{ChannelMean}(\mathbf{H}_{out}\mathbf{W}_s\mathbf{W}_{ano}) \in \mathbb{R}^{T}$, where $\mathbf{W}_s \in \mathbb{R}^{D_v \times D_v}$ is the shared embedding projection, $\mathbf{W}_{rec} \in \mathbb{R}^{D_v \times W}$ and $\mathbf{W}_{ano} \in \mathbb{R}^{D_v \times W}$ are the reconstruction and anomaly projections, respectively. The reconstruction head predicts masked portions of the input series. This head should be understood as a representation-learning pretext task rather than as the detector itself: we critique reconstruction error as an inference-time anomaly score, not masked reconstruction as a training signal. To foster robust contextual learning, we employ a **patched masking strategy** (15% ratio), preventing information leakage from adjacent timestamps. This design encourages the model to infer missing patterns by aggregating context, thereby establishing the "normality reference" required for the RCD mechanism.

The rationale of this dual-head design stems from the nature of supervision. The BCE signal for anomaly labels defines the task-specific decision boundary, but it is relatively sparse and can be difficult to optimize from scratch. The masked reconstruction loss provides dense token-level supervision, encouraging the encoder to learn temporal dynamics and

dependencies that serve as an optimization scaffold for the anomaly objective. We optimize the model using a joint loss function:

$$\mathcal{L} = \lambda\|\mathbf{M} \odot (\mathbf{X}_{rec} - \mathbf{X})\|_2^2 + \text{BCE}(\sigma(\mathbf{X}_{ano}), \mathbf{y}),$$

where $\mathbf{M}$ is the binary mask.

It is vital to note that this reconstruction branch is functionally distinct from the detection logic. During inference, **the Reconstruction Head is discarded completely**, and no masking is applied. The model relies exclusively on the learned latent representations to classify anomalies via the Anomaly Head. This separates our paradigm from reconstruction-error methods: we do not measure *how well* the model reconstructs the input at test time; rather, we use reconstruction during training to learn dense temporal representations that make anomaly-head optimization more stable. A controlled objective/inference-rule ablation is provided in Table 3, with an appendix pointer in Appendix E.5. The results show that reconstruction is useful as a training-time scaffold, while the anomaly head and joint representation learning are essential for strong inference-time detection.

### 2.3. Context-Dependent Data Generation

We design a synthetic engine to generate multivariate time series without fixed semantic coupling, mitigating the model's reliance on memorized global priors. By ensuring that "normality" is context-specific and defined by the sequence itself, this synthetic-data design encourages models to recognize anomalies primarily relative to the sequence context. The pipeline proceeds in three hierarchical stages designed to *establish* and then *violate* context: first, **constructing univariate dominant patterns** through an additive model with trend, seasonality, and noise (Stage 1); next, **enforcing multivariate consistency** via causal dependencies to establish cross-variate context (Stage 2); and finally, **violating these established contexts** by injecting structural and causal anomalies (Stage 3). A summary is shown in Fig. 4, with full details in Appendix B.

**Stage 1: Establishing Univariate Dominant Patterns.** To encourage the model to derive normality from the sequence context rather than through global memorization, we first generate univariate baselines with highly randomized structural parameters. For each channel, the normal context is constructed via an additive model: $x_{\text{base}}(t) = T(t) + S(t) + \varepsilon(t)$, $t = 0, 1, \ldots, n-1$. Here, the **Trend** $T(t)$ combines deterministic and stochastic drifts to simulate diverse evolving baselines. Crucially, the **Seasonality** $S(t) = \sum_{k=1}^{K} A_k w_k(2\pi f_k t + \varphi_k; \theta_k)$ serves as the primary "dominant pattern" and provides the reference against which deviations are identified. By randomly sampling the waveform atoms $w_k$ (e.g., sinusoid, square, wavelet), frequencies

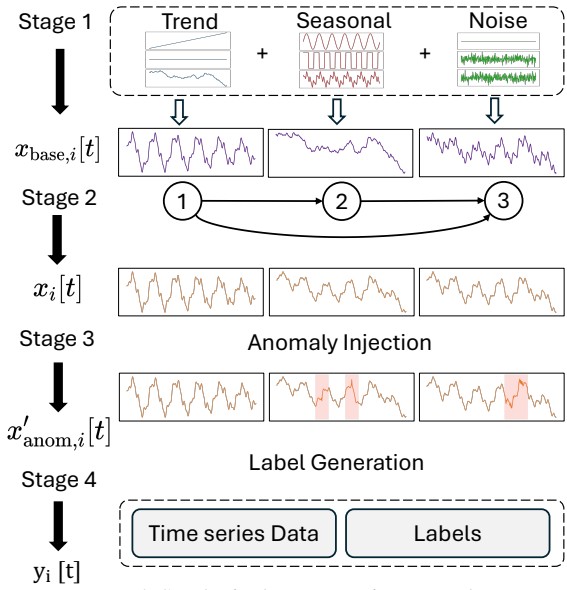

Stage 1

Stage 2

Stage 3

Anomaly Injection

Stage 4

Label Generation

*Figure 4.* Synthetic data generation procedure.

$f_k$, and phases $\varphi_k$ for each sample, we prevent any single periodic structure from being universally applicable across all sequences. Finally, the **Noise** $\varepsilon(t)$ introduces local volatility (e.g., bursty variance $\sigma^2(t)$), requiring the model to distinguish inherent stochastic volatility from genuine anomalies. Full construction details are provided in Appendix B.2 and Appendix B.3.

**Stage 2: Enforcing Multivariate Consistency**   To establish relational context across channels, we integrate the independent univariate patterns into a structurally coupled system. This step is critical for promoting cross-variate attention, as "normality" is now defined by the consistency between interacting nodes. We first sample a Directed Acyclic Graph (DAG) $G = (V, E)$ to define the topological structure of dependencies. To impose precise temporal constraints, we employ a latent causal process $z_i$ governed by discretized ARX dynamics: $z_i[t] = a_i z_i[t-1] + \sum_{j \in P(i)} b_{ij} x_j[t - \ell_{ij}] + c_i$, $\quad |a_i| \le 0.8$, where the parent set $P(i)$ and specific lags $\ell_{ij}$ create lagged dependencies essential for the model to capture. Finally, the observed signal is generated by mixing the univariate baseline with this causal process: $x_i[t] = (1 - \alpha_i) x_{\text{base},i}(t) + \alpha_i z_i[t]$, $\quad \alpha_i \in [0, 1]$. This mechanism couples the channels, making a channel's behavior conditional upon the states of its parents, thereby establishing a structured multivariate ground truth for the RCD task.

**Stage 3: Violating Established Contexts**   The final step involves generating anomalies not through random corruption, but by disrupting the contextual patterns established in previous stages. We introduce two classes of injections designed to necessitate the RCD mechanism:

**(1) Violating Temporal Context (Structural & Local Anomalies)** We apply a broad spectrum of perturbations, ranging from **local morphological changes** (adding $\Delta(t)$ to introduce shape-based outliers) to more challenging **structural violations**. Specifically, for the latter, we intervene on the seasonality component $S(t)$, replacing it with $S'(t)$ to induce frequency shifts, phase jumps, or pattern replacements. Since these structural anomalies often morphologically resemble **legitimate normal patterns** found in other generated series, they are *difficult to distinguish based on shape alone*. Effective identification relies on contrasting the current window against its own extended temporal context, thereby driving the model to *infer context-level normality rather than relying on static global priors*. We design over 20 anomaly types (detailed in Appendix B.4) to cover these violations.

**(2) Violating Relational Context (Endogenous Injections)** To challenge the model's capacity for capturing multivariate dependencies, we employ an **endogenous injection mechanism**. Rather than simply overwriting the observed signal (Exogenous), we perturb the latent baseline $x_{\text{base},i}$ of a variable prior to the mixing process. The induced deviation consequently propagates to other correlated variables via the system dynamics defined in Stage 2. This process generates anomalies that preserve structural coherence but violate the **inter-variable consistency** dictated by the multivariate context. Detecting such anomalies hinges on capturing whether the joint behavior of variables remains consistent, facilitating the learning of cross-variate relational discrepancies without the need for explicit causal modeling.

**Stage 4: Label Generation**   We generate token-level binary labels. For exogenous injections, positive labels correspond to the intervention window. For endogenous injections, we label the root-cause window and extend it to descendant channels based on their causal lags, as implied by the DAG and ARX lags $\{\ell_{ij}\}$. Sequence lengths, DAG sparsity, ARX coefficients, and signal regimes (trend, seasonality, noise) are sampled from configurable priors (Appx. B.1), generating rich, interpretable dynamics designed for zero-shot learning.

## 3. Experiments

To validate the effectiveness of TimeRCD, we design a comprehensive evaluation to answer three research questions: **RQ1**: How well does TimeRCD perform in strict zero-shot anomaly detection compared with existing time-series foundation models and with full-shot, dataset-specific baselines? **RQ2**: What is the impact of large context windows on contextual anomaly detection and sensitivity to window size? **RQ3**: How do our synthetic generator and anomaly design affect performance, and how does accuracy scale with

pre-training data size?

## 3.1. Experimental Settings

**Datasets** Our evaluation is conducted on a comprehensive suite of 16 public time-series anomaly detection datasets, covering a wide range of real-world and synthetic scenarios. Details about the benchmark datasets can be found in Appendix D.1.

**Baselines** We benchmark TimeRCD against methods from two primary settings: (1) **Zero-shot models**, which include our approach and other foundation models (DADA† (Shentu et al., 2024), MOMENT† (Goswami et al., 2024), TimesFM (Das et al., 2024), Chronos (Ansari et al., 2024), Time MOE (Shi et al., 2024)). We also include **MovingVar.**, a simple moving-window variance detector, to test whether a lightweight statistical baseline can explain the observed gains. (2) **Full-shot models**, which are fitted on a per-dataset basis. This category includes deep learning methods (TranAD (Tuli et al., 2022), USAD (Audibert et al., 2020), OmniAnomaly (Su et al., 2019), Sub-PCA (Liu & Paparrizos, 2024), DCdetector (Yang et al., 2023), TF-MAE (Fang et al., 2024)) and classical statistical algorithms (LOF (Breunig et al., 2000), IForest (Liu et al., 2008)). Note that models marked with (†) were excluded where necessary due to potential data leakage under the zero-shot setting (Appendix D.3). Additional details about all baselines can be found in Appendix D.2.

**Evaluation Protocol** We evaluate model performance using four standard metrics: Affiliation-F1, F1-T, Standard-F1, and VUS-PR. More details about the metrics are shown in Appendix D.4. TimeRCD uses a single fixed context window of 5000 for every dataset.

## 3.2. Main Results: TSAD Accuracy (RQ1)

Our evaluation includes two comparisons: a direct **zero-shot** test against foundation models, and a **full-shot** test against baselines trained on target data. We stress that **TimeRCD** is strictly zero-shot in all settings, testing true out-of-the-box performance under the fixed 5000 context-window protocol. Results are shown in Table 1. In zero-shot comparisons, TimeRCD ranks first in **38** and second in **14** of the 64 evaluation cases (16 datasets × 4 metrics). Even against full-shot baselines with access to target data, TimeRCD is highly competitive, ranking first in **35** and second in **3** cases. This result highlights the effectiveness and cross-dataset generalization of our pre-training framework. The MovingVar. baseline is competitive on several datasets and metrics, especially when variance shifts are directly aligned with the anomaly definition. Nevertheless, TimeRCD is stronger overall and achieves substantially more first-place results, including broad gains on VUS-PR. This comparison suggests that the performance cannot be reduced to a simple variance detector, while also confirming that simple statistical baselines remain meaningful competitors in TSAD.

## 3.3. RCD Strategy Efficiency (RQ2)

**Qualitative Analysis of Contextual Understanding** A key architectural feature of TimeRCD is its ability to process long context windows, allowing it to learn complex temporal dependencies. Many existing zero-shot methods, particularly those based on reconstruction with small look-up windows, are effective at detecting abrupt **point anomalies**—short-term deviations from an immediate pattern (the left two panels in Fig. 5). However, these models often fail on subtle **contextual anomalies**, where the anomalous behavior is a deviation from a long-term pattern (the right two panels in Fig. 5). Their limited context prevents them from distinguishing normal long-term variations from true anomalous segments. In contrast, as our qualitative results in Fig. 5 show, TimeRCD's ability to view the entire series allows it to learn the complex relationships between distant points.

**Quantitative Analysis of Contextual Understanding** We create specialized, unseen datasets containing either purely point or contextual anomalies, ensuring a fair zero-shot evaluation (details in Appendix E.2). The results are in Fig. 6. While TimeRCD's performance on point anomalies is highly competitive with other top zero-shot models, it is substantially stronger on contextual anomalies. On this task, our model achieves a Standard-F1 score of 0.827, whereas competing zero-shot models degrade substantially on this split. This performance disparity provides strong evidence that TimeRCD's ability to leverage long-range context is a key capability, allowing it to detect complex deviations that are challenging for methods with a more limited contextual view.

**Impact of Context Window Size** TimeRCD's ability to process variable context lengths is a core architectural feature. To analyze its impact, we evaluate performance with input window sizes from 1k to 13k. As shown in Fig. 7, the results confirm that the optimal context length is task-dependent. For datasets with long-term patterns like **UCR**, **Power**, **SMAP**, and **SMD**, performance generally improves with a larger window, as this allows the model to establish a more robust baseline of "normal" behavior. Conversely, on inherently short series like **YAHOO** (Fig. 7(c)), performance remains flat, as the series length itself becomes the effective context limit. Practically, smaller windows are often preferable when anomalies are highly local or the series has weak periodic structure, while larger windows are more helpful when normality depends on long-range context, periodicity, or stronger contextual structure. Thus, the fixed

*Table 1.* Performance of **TimeRCD** against zero-shot and full-shot baselines. TimeRCD operates in a strictly zero-shot capacity in all comparisons. Best results are highlighted in red; second-best results are shown in blue. Asterisked (*) results are excluded from ranking due to potential data leakage.

| Metric | Model | Univariate Datasets | | | | | | | | | | | Multivariate Datasets | | | | | Total | Total |
|---|---|---|---|---|---|---|---|---|---|---|---|---|---|---|---|---|---|---|---|
| | | IOPS | MGAB | NAB | NEK | Power | SED | Stock | TODS | UCR | WSD | YAHOO | MSL | PSM | SMAP | SMD | SWaT | 1st | 2nd |
| **Zero-Shot Models** | | | | | | | | | | | | | | | | | | | |
| Affiliation-F | TimeRCD | 86.98 | 74.54 | 93.81 | 83.95 | 85.02 | 88.99 | 41.31 | 84.86 | 84.34 | 91.00 | 95.94 | 84.66 | 82.31 | 86.01 | 91.75 | 69.43 | 07 | 04 |
| | DADA† | 89.37* | 67.66* | 86.56 | 95.40 | 69.79 | 65.18 | 98.77 | 76.89 | 72.21 | 93.92 | 92.20* | 76.57 | 81.27 | 76.92 | 83.74 | 76.18 | 03 | 03 |
| | MOMENT† | 87.54* | 66.76* | 90.45* | 92.26 | 75.97 | 59.13 | 45.26 | 59.76 | 75.77 | 95.39 | 79.99* | 74.55* | 65.79 | 77.42* | 74.00* | 70.17 | 01 | 01 |
| | TimesFM | 81.88 | 66.95 | 79.73 | 90.49 | 68.85 | 67.14 | 97.53 | 89.08 | 74.67 | 78.97 | 79.70 | 20.35 | 71.24 | 45.44 | 62.85 | 44.37 | 00 | 02 |
| | Chronos | 90.12 | 67.89 | 86.66 | 93.63 | 69.72 | 67.89 | 96.85 | 91.96 | 74.35 | 90.98 | 96.34 | 75.52 | 70.88 | 72.22 | 75.31 | 70.43 | 03 | 03 |
| | Time MOE | 76.34 | 67.23 | 80.51 | 80.50 | 71.19 | 60.98 | 63.28 | 54.68 | 73.56 | 80.25 | 69.70 | 69.85 | 54.74 | 74.38 | 69.97 | 64.37 | 00 | 00 |
| | MovingVar. | 83.90 | 67.56 | 91.50 | 77.81 | 74.77 | 67.14 | 67.96 | 71.57 | 78.44 | 89.13 | 78.00 | 89.43 | 73.91 | 88.14 | 81.33 | 75.55 | 02 | 03 |
| F1-T | TimeRCD | 45.39 | 9.05 | 50.49 | 76.91 | 23.05 | 60.59 | 18.38 | 68.46 | 41.18 | 44.13 | 79.14 | 43.67 | 43.86 | 35.82 | 55.96 | 35.47 | 11 | 03 |
| | DADA† | 42.50* | 0.91* | 37.24 | 47.98 | 19.80 | 9.56 | 95.49 | 35.18 | 7.22 | 48.46 | 79.52* | 34.58 | 31.84 | 30.42 | 40.80 | 35.13 | 02 | 04 |
| | MOMENT† | 33.15* | 0.80* | 52.27* | 63.66 | 19.91 | 9.54 | 18.04 | 17.47 | 13.02 | 41.98 | 11.69* | 25.97* | 27.77 | 17.93* | 28.68* | 28.76 | 00 | 02 |
| | TimesFM | 48.95 | 0.93 | 36.74 | 36.63 | 19.80 | 9.58 | 88.94 | 51.13 | 10.78 | 41.38 | 83.46 | 7.83 | 25.42 | 11.64 | 18.65 | 21.39 | 02 | 00 |
| | Chronos | 45.45 | 1.10 | 36.10 | 33.16 | 19.90 | 13.18 | 89.30 | 53.90 | 10.88 | 39.82 | 79.00 | 15.59 | 25.42 | 11.72 | 17.32 | 28.88 | 00 | 04 |
| | Time MOE | 25.95 | 0.63 | 38.70 | 15.78 | 19.85 | 17.73 | 34.13 | 20.91 | 8.29 | 22.60 | 37.11 | 23.92 | 26.82 | 14.22 | 19.90 | 30.11 | 00 | 01 |
| | MovingVar. | 22.53 | 0.85 | 56.66 | 55.72 | 19.93 | 9.59 | 17.53 | 16.43 | 12.85 | 22.01 | 13.89 | 41.08 | 26.13 | 24.13 | 32.67 | 27.11 | 01 | 02 |
| Standard-F1 | TimeRCD | 47.28 | 8.63 | 42.66 | 68.64 | 23.12 | 60.85 | 19.56 | 70.02 | 33.47 | 41.04 | 89.05 | 34.48 | 28.25 | 25.36 | 53.82 | 34.44 | 10 | 04 |
| | DADA† | 32.76* | 0.80* | 26.91 | 48.24 | 15.99 | 2.69 | 95.59 | 28.18 | 3.36 | 45.06 | 79.30* | 22.13 | 24.07 | 26.75 | 34.98 | 34.78 | 04 | 00 |
| | MOMENT† | 30.69* | 0.67* | 44.75* | 63.85 | 16.39 | 3.36 | 19.38 | 14.64 | 9.00 | 41.42 | 10.54* | 14.43* | 23.83 | 12.92* | 29.78* | 21.30 | 00 | 02 |
| | TimesFM | 34.28 | 0.83 | 26.46 | 38.15 | 16.73 | 2.96 | 89.13 | 40.08 | 7.86 | 38.50 | 84.44 | 5.75 | 22.18 | 10.46 | 18.65 | 22.84 | 00 | 02 |
| | Chronos | 32.69 | 0.99 | 26.22 | 33.54 | 17.47 | 8.74 | 89.41 | 40.52 | 8.21 | 34.58 | 78.89 | 11.63 | 22.27 | 9.62 | 17.50 | 24.03 | 00 | 03 |
| | Time MOE | 26.52 | 0.45 | 26.20 | 11.47 | 12.16 | 17.73 | 34.32 | 16.38 | 4.09 | 20.09 | 27.50 | 12.85 | 24.80 | 9.01 | 21.62 | 23.58 | 00 | 01 |
| | MovingVar. | 22.74 | 0.74 | 51.37 | 49.19 | 19.94 | 9.51 | 19.01 | 16.10 | 9.51 | 22.57 | 11.62 | 35.43 | 26.10 | 23.66 | 38.43 | 26.86 | 02 | 04 |
| VUS-PR | TimeRCD | 42.98 | 3.09 | 44.65 | 71.13 | 14.63 | 69.86 | 72.73 | 91.67 | 27.48 | 29.95 | 87.88 | 23.41 | 21.69 | 19.68 | 46.70 | 18.47 | 10 | 03 |
| | DADA† | 24.97* | 0.57* | 24.73 | 46.85 | 10.61 | 6.42 | 99.51 | 64.83 | 2.94 | 33.42 | 70.74* | 12.74 | 17.17 | 20.02 | 25.98 | 21.13 | 03 | 03 |
| | MOMENT† | 37.35* | 0.56* | 45.38* | 67.74 | 10.50 | 4.31 | 76.97 | 56.45 | 6.17 | 55.26 | 30.81* | 9.32* | 16.48 | 8.97* | 15.96* | 21.13 | 01 | 01 |
| | TimesFM | 19.56 | 0.58 | 24.01 | 35.02 | 10.44 | 6.13 | 98.39 | 72.89 | 6.03 | 21.57 | 86.78 | 11.84 | 14.76 | 16.95 | 13.02 | 19.43 | 00 | 04 |
| | Chronos | 19.00 | 0.60 | 23.76 | 31.80 | 10.95 | 8.65 | 97.49 | 70.66 | 6.56 | 18.81 | 83.54 | 8.25 | 14.61 | 5.18 | 10.22 | 16.44 | 00 | 01 |
| | Time MOE | 16.63 | 0.52 | 22.62 | 19.76 | 9.34 | 10.87 | 74.78 | 48.78 | 2.10 | 10.93 | 20.90 | 7.82 | 15.68 | 4.98 | 11.12 | 16.20 | 00 | 01 |
| | MovingVar. | 22.47 | 0.44 | 48.75 | 38.81 | 10.70 | 5.17 | 73.99 | 51.58 | 6.90 | 22.83 | 19.85 | 25.02 | 16.45 | 12.73 | 21.43 | 14.49 | 02 | 02 |
| **TimeRCD Grand Total (Zero-Shot)** | | | | | | | | | | | | | | | | | | 38 | 14 |
| **Full-Shot Models** | | | | | | | | | | | | | | | | | | | |
| Affiliation-F | TimeRCD | 86.98 | 74.54 | 93.81 | 83.95 | 85.02 | 88.99 | 41.31 | 84.86 | 84.34 | 91.00 | 95.94 | 84.66 | 82.31 | 86.01 | 91.75 | 69.43 | 11 | 01 |
| | TranAD | 83.19 | 67.28 | 90.28 | 85.02 | 71.56 | 61.03 | 57.94 | 52.76 | 73.31 | 84.34 | 76.08 | 79.91 | 73.83 | 87.39 | 92.20 | 75.37 | 01 | 03 |
| | USAD | 71.08 | 67.81 | 91.54 | 71.13 | 76.48 | 55.60 | 35.92 | 47.90 | 76.00 | 65.10 | 53.05 | 81.86 | 57.86 | 87.25 | 85.09 | 75.06 | 00 | 00 |
| | OmniAnomaly | 80.32 | 67.35 | 92.35 | 86.30 | 78.16 | 61.26 | 75.24 | 50.73 | 73.53 | 78.02 | 71.31 | 83.15 | 58.17 | 91.38 | 85.82 | 73.39 | 02 | 03 |
| | LOF | 81.06 | 68.44 | 75.75 | 84.74 | 66.76 | 63.85 | 69.74 | 60.58 | 73.53 | 81.29 | 75.63 | 84.35 | 61.98 | 63.32 | 64.13 | 56.34 | 00 | 01 |
| | IForest | 52.81 | 68.82 | 39.84 | 71.15 | 0.00 | 70.09 | 0.06 | 44.17 | 50.56 | 41.24 | 33.30 | 63.36 | 63.78 | 59.96 | 69.71 | 0.00 | 00 | 01 |
| | Sub-PCA | 75.39 | 66.90 | 89.29 | 97.10 | 71.37 | 67.14 | 70.63 | 72.75 | 76.66 | 76.45 | 75.85 | 84.25 | 71.49 | 90.08 | 85.80 | 76.29 | 02 | 04 |
| | DCdetector | 71.83 | 67.91 | 72.21 | 62.31 | 69.75 | 72.20 | 55.79 | 57.81 | 70.18 | 72.79 | 67.77 | 67.74 | 67.32 | 67.10 | 69.55 | 71.07 | 00 | 01 |
| | TFMAE | 78.25 | 67.50 | 75.99 | 76.91 | 70.30 | 68.17 | 56.39 | 62.83 | 70.60 | 80.25 | 76.87 | 75.70 | 70.07 | 75.36 | 70.85 | 75.72 | 00 | 02 |
| F1-T | TimeRCD | 45.39 | 9.05 | 50.49 | 76.91 | 23.05 | 60.59 | 18.38 | 68.46 | 41.18 | 44.13 | 79.14 | 43.67 | 43.86 | 35.82 | 55.96 | 35.47 | 08 | 01 |
| | TranAD | 22.63 | 1.65 | 37.28 | 69.97 | 22.36 | 9.57 | 16.73 | 13.51 | 7.75 | 20.94 | 8.41 | 39.42 | 25.49 | 29.12 | 37.98 | 49.58 | 00 | 01 |
| | USAD | 20.99 | 4.07 | 61.46 | 70.64 | 28.23 | 9.54 | 16.86 | 20.85 | 14.63 | 14.18 | 9.35 | 48.71 | 25.86 | 43.94 | 50.41 | 50.41 | 03 | 01 |
| | OmniAnomaly | 51.17 | 1.61 | 40.09 | 82.20 | 23.48 | 9.68 | 36.22 | 14.33 | 8.47 | 34.79 | 24.16 | 49.36 | 30.42 | 46.63 | 51.84 | 46.64 | 03 | 05 |
| | LOF | 27.97 | 1.15 | 35.76 | 63.57 | 19.80 | 9.60 | 66.14 | 31.63 | 8.31 | 24.38 | 55.93 | 38.97 | 25.58 | 21.81 | 10.13 | 30.62 | 01 | 02 |
| | IForest | 7.64 | 0.84 | 21.44 | 65.56 | 0.00 | 9.54 | 1.10 | 11.06 | 6.36 | 4.28 | 4.90 | 20.73 | 25.39 | 14.32 | 16.20 | 0.00 | 00 | 00 |
| | Sub-PCA | 32.75 | 0.98 | 54.15 | 84.18 | 20.30 | 9.54 | 20.10 | 18.40 | 18.90 | 24.62 | 11.57 | 49.02 | 30.38 | 44.31 | 51.87 | 46.65 | 01 | 05 |
| | DCdetector | 6.61 | 1.32 | 32.72 | 29.21 | 21.13 | 10.53 | 16.07 | 16.40 | 6.62 | 7.32 | 6.81 | 23.24 | 25.34 | 15.73 | 9.47 | 28.64 | 00 | 00 |
| | TFMAE | 19.41 | 1.07 | 33.04 | 31.11 | 20.18 | 11.82 | 22.15 | 16.48 | 5.90 | 19.11 | 6.87 | 15.68 | 25.26 | 19.39 | 10.13 | 28.46 | 00 | 02 |
| Standard-F1 | TimeRCD | 47.28 | 8.63 | 42.66 | 68.64 | 23.12 | 60.85 | 19.56 | 70.02 | 33.47 | 41.04 | 89.05 | 34.48 | 28.25 | 25.36 | 53.82 | 34.44 | 07 | 00 |
| | TranAD | 34.85 | 1.46 | 27.33 | 60.36 | 22.36 | 2.63 | 16.23 | 11.94 | 4.40 | 20.23 | 5.70 | 29.60 | 25.63 | 25.11 | 43.99 | 61.86 | 00 | 01 |
| | USAD | 30.66 | 3.89 | 56.15 | 62.91 | 28.24 | 3.41 | 17.99 | 23.87 | 10.74 | 12.90 | 7.21 | 38.71 | 28.24 | 38.66 | 53.06 | 62.82 | 03 | 01 |
| | OmniAnomaly | 47.05 | 1.44 | 28.81 | 74.03 | 23.50 | 0.43 | 38.59 | 12.65 | 5.11 | 29.57 | 21.40 | 39.10 | 30.43 | 40.50 | 57.06 | 55.93 | 03 | 06 |
| | LOF | 30.28 | 1.05 | 24.04 | 56.92 | 12.18 | 4.11 | 66.20 | 25.77 | 4.70 | 22.62 | 48.95 | 30.65 | 18.80 | 18.70 | 8.41 | 29.08 | 01 | 02 |
| | IForest | 8.37 | 0.73 | 29.41 | 58.10 | 19.77 | 3.81 | 16.91 | 13.35 | 4.09 | 2.07 | 3.20 | 14.68 | 24.15 | 13.61 | 16.89 | 26.76 | 00 | 00 |
| | Sub-PCA | 33.96 | 0.83 | 46.71 | 85.43 | 16.05 | 9.56 | 21.77 | 18.72 | 15.12 | 24.74 | 11.06 | 38.29 | 30.26 | 38.90 | 57.22 | 55.98 | 02 | 04 |
| | DCdetector | 5.19 | 1.21 | 24.02 | 17.37 | 21.10 | 10.54 | 16.97 | 17.85 | 3.18 | 4.64 | 4.14 | 14.08 | 25.33 | 10.67 | 8.99 | 27.02 | 00 | 00 |
| | TFMAE | 9.48 | 0.97 | 23.78 | 19.74 | 20.14 | 11.87 | 23.32 | 14.88 | 2.83 | 15.53 | 20.50 | 15.68 | 25.39 | 12.58 | 9.16 | 27.08 | 00 | 01 |
| VUS-PR | TimeRCD | 42.98 | 3.09 | 44.65 | 71.13 | 14.63 | 69.86 | 72.73 | 91.67 | 27.48 | 29.95 | 87.88 | 23.41 | 21.69 | 19.68 | 46.70 | 18.47 | 09 | 01 |
| | TranAD | 21.61 | 0.64 | 24.82 | 61.63 | 10.54 | 5.75 | 78.08 | 47.33 | 2.25 | 12.29 | 25.78 | 14.78 | 16.49 | 13.37 | 28.34 | 47.37 | 01 | 00 |
| | USAD | 16.58 | 0.75 | 55.03 | 58.53 | 18.68 | 4.37 | 74.53 | 56.36 | 8.85 | 10.00 | 14.15 | 29.95 | 17.59 | 26.37 | 34.53 | 44.73 | 02 | 03 |
| | OmniAnomaly | 25.35 | 0.64 | 27.17 | 74.51 | 14.32 | 6.20 | 91.29 | 45.55 | 2.40 | 16.37 | 29.26 | 31.57 | 18.58 | 28.07 | 37.44 | 42.97 | 03 | 03 |
| | LOF | 19.43 | 0.57 | 21.18 | 58.52 | 9.31 | 6.83 | 83.07 | 49.14 | 2.39 | 12.85 | 41.37 | 24.67 | 13.58 | 10.59 | 4.40 | 14.50 | 00 | 02 |
| | IForest | 8.59 | 0.62 | 23.57 | 56.50 | 11.56 | 7.71 | 70.99 | 46.62 | 2.88 | 2.06 | 10.47 | 11.29 | 15.85 | 7.55 | 8.88 | 15.49 | 00 | 00 |
| | Sub-PCA | 23.02 | 0.60 | 46.08 | 88.91 | 10.49 | 3.72 | 80.86 | 54.16 | 12.92 | 16.41 | 21.57 | 31.43 | 18.52 | 26.42 | 37.50 | 43.02 | 01 | 06 |
| | DCdetector | 5.83 | 0.59 | 16.60 | 14.03 | 12.32 | 9.37 | 74.16 | 46.66 | 1.53 | 3.23 | 10.17 | 7.01 | 14.49 | 4.21 | 4.66 | 15.04 | 00 | 00 |
| | TFMAE | 5.32 | 0.64 | 15.68 | 17.81 | 11.90 | 9.55 | 73.54 | 48.79 | 2.57 | 5.36 | 25.93 | 8.25 | 14.22 | 5.76 | 4.77 | 15.38 | 00 | 01 |
| **TimeRCD Grand Total (Full-Shot)** | | | | | | | | | | | | | | | | | | 35 | 03 |

5000 window used in Table 1 is a practical zero-shot default, not a claim that context length is irrelevant. A detailed breakdown for all datasets is available in Appendix E.3.

### 3.4. Synthetic Data Efficiency (RQ3)

**Ablation Study on Pre-training Data** To validate our data generation framework, we train the same TimeRCD architecture with identical hyperparameters and epochs on three 350M-point datasets: (1) our synthetic data with in-context anomaly injection, (2) the same synthetic series but with DADA-injected anomalies (Shentu et al., 2024), and (3) real-world data (Godahewa et al.) augmented with DADA. Table 2 shows weighted averages over all univariate benchmarks. Models trained on augmented real data performed far worse, confirming the necessity of a high-quality synthetic curriculum. Comparing the two synthetic variants reveals a key trade-off: although DADA injection achieves similar Affiliation-F and slightly higher VUS-PR, it causes sharp

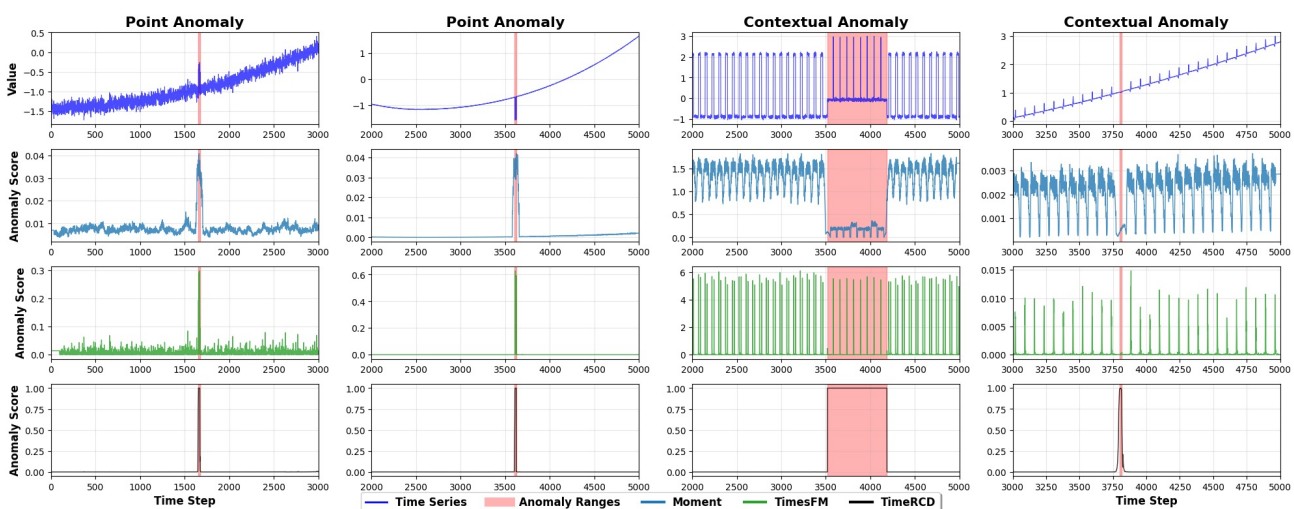

*Figure 5.* Qualitative comparison of anomaly scores for point and contextual anomalies.

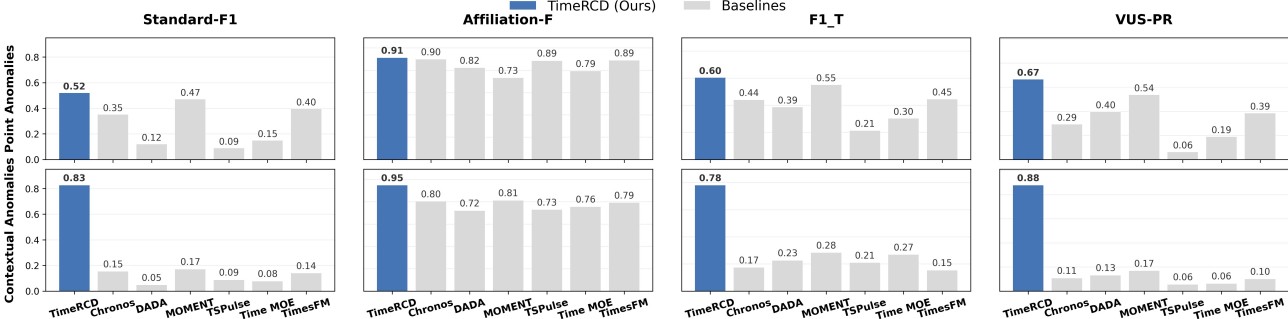

*Figure 6.* Comparison on specialized datasets containing either point or contextual anomalies.

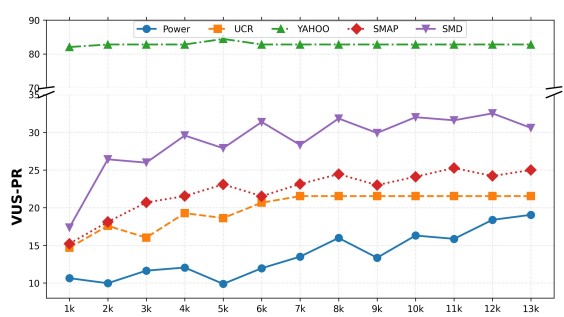

*Figure 7.* The effect of input window size on datasets (Power, UCR, YAHOO, SMAP, and SMD) with sizes ranging from 1k to 13k.

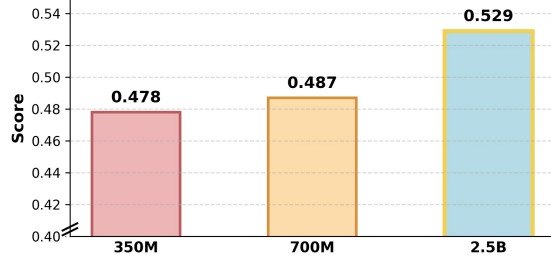

*Figure 8.* Empirical scaling trend for average VUS-PR across our benchmarks when increasing the number of training observations (350M, 700M, 2.5B).

*Table 2.* Weighted average performance across all univariate benchmarks.

| Dataset | Aff-F | F1-T | Std-F1 | VUS-PR |
|---|---|---|---|---|
| Our Synthetic Data | **0.878** | **0.569** | **0.523** | 0.478 |
| Our Synthetic + DADA Injection | **0.878** | 0.505 | 0.462 | **0.487** |
| Real-world Data + DADA Injection | 0.716 | 0.073 | 0.062 | 0.102 |

drops in finer-grained metrics (F1-T decreases by 6.4 percentage points and Standard-F1 by 6.1 percentage points). This indicates that our in-context injection generates more challenging and robust training signals.

**Dataset Scaling**   To investigate the effect of pre-training data scale on performance, we train TimeRCD on increasingly larger subsets of our synthetic dataset: 350M, 700M, and the full 2.5B data points. The results, shown as a weighted average across our benchmark datasets in Fig. 8, demonstrate a clear positive trend in average VUS-PR as

the amount of pre-training data increases. Full per-metric results are provided in Appendix E.4.

### 3.5. Controlled Ablation on Objective and Inference Rule

To better isolate the roles of synthetic data, reconstruction, and the anomaly head, we perform a controlled comparison on UCR using the same backbone, tokenization, synthetic corpus, and comparable training budget while varying only the training objective and inference rule. The results are shown in Table 3.

*Table 3.* Controlled UCR ablation with the same backbone, tokenization, and synthetic corpus.

| Exp. | Training Objective | Inference Rule | VUS-PR |
|---|---|---|---|
| Exp1 | Reconstruction + Anomaly | Anomaly head | **33.57** |
| Exp2 | Reconstruction + Anomaly | Reconstruction error | 4.08 |
| Exp3 | Reconstruction only | Reconstruction error | 3.35 |
| Exp4 | Anomaly only | Anomaly head | 5.90 |
| Exp5 | Reconstruction pretraining, frozen encoder, then anomaly-head training | Anomaly head | 26.11 |

These comparisons clarify the role of each component. Exp1 vs. Exp3 shows that reconstruction-only training on the same synthetic corpus does not recover the full method, so the gains are not explained by data scaling alone. Exp1 vs. Exp2 shows that, even under identical joint training, replacing the anomaly head with reconstruction error at inference causes a large drop, supporting our distinction between useful training-time reconstruction and weak test-time reconstruction-error scoring. Exp4 vs. Exp5 shows that training the anomaly head from scratch is difficult under sparse BCE supervision, whereas a reconstruction-pretrained encoder substantially stabilizes the anomaly objective. Exp3 vs. Exp5 further shows that adding an anomaly head on top of a learned representation is much stronger than using reconstruction error alone. Finally, Exp1 remains better than Exp5, indicating that jointly adapting the representation to the anomaly objective is also important.

## 4. Conclusion

In conclusion, TimeRCD addresses limitations of reconstruction-error inference by introducing Relative Context Discrepancy as a context-relative pre-training paradigm for zero-shot TSAD. Its conceptual simplicity and empirical results suggest several directions for future research. As a current limitation and future direction, our implementation relies on a standard Transformer backbone; exploring architectures designed to capture RCD more efficiently remains an important direction for improving efficiency.

**Limitations.** Several limitations remain important for interpreting the current results. First, the context-relative design can fail when a large inference window contains many anomalous points: on datasets such as Stock, frequent outliers inside the same window may be partially absorbed into the local reference pattern, weakening the anomaly signal. Second, the fixed context window of 5000 used in main results is a practical zero-shot default, not evidence that context length is unimportant; some datasets remain

sensitive to the window length. Third, reconstruction has a nuanced role in TimeRCD: it is not used as the test-time anomaly score, but masked reconstruction provides a dense representation-learning and optimization scaffold for the sparse anomaly objective. A more rigorous theoretical account of this interaction is an open problem.

## Acknowledgements

This work was supported in part by NSFC Grant 72271138, Tsinghua-NUS Joint Funding 20243080039, and Tsinghua-Huawei Collaboration Project 20252001894. We thank Zixinzhou Yang for helpful contributions to the dataset. The code is available at `https://github.com/thu-sail-lab/Time-RCD`.

## Impact Statement

This work may reduce the amount of target-domain data required for time series anomaly detection, which could benefit applications such as industrial monitoring, cloud operations, and healthcare. Because anomaly detection systems can affect high-stakes decisions, deployments should include human oversight, domain-specific validation, and careful monitoring of false positives and false negatives. Our use of synthetic pre-training data reduces direct exposure to private downstream data, but downstream deployment must still follow applicable privacy and security requirements.

**Use of LLMs.** Large language models were used only to refine the writing.

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

# A. Related Work

**State-of-the-art TSAD algorithm** Traditional unsupervised TSAD operates by training models on domain-specific normal data to identify deviations in unseen test sets. Within this paradigm, two primary approaches have been dominant. Reconstruction-based methods identify anomalies by quantifying the error when attempting to rebuild an input signal, with higher errors indicating anomalous behavior (Audibert et al., 2020; Su et al., 2019; Zhang et al., 2019; Tuli et al., 2022; Nam et al., 2024). In parallel, discrepancy-based methods learn discriminative representations by maximizing the distinction between normal and anomalous patterns or distributions (Xu et al., 2021; Yang et al., 2023; Shen et al., 2020; Darban et al., 2025; Lan et al., 2025; Wilson et al., 2023). However, the reliance of these methods on in-domain normal data presents a fundamental barrier to generalization, leading to significant performance degradation in zero-shot scenarios where the model encounters data from new, unseen domains. While the recent emergence of foundation models aims to overcome this transferability issue, many approaches remain anchored in reconstruction-error inference (Shentu et al., 2024; Gao et al., 2024; Goswami et al., 2024; Liu et al., 2025; Yoon et al., 2025), which can suffer from an **objective mismatch**. To our knowledge, this work is among the first foundation-model approaches for zero-shot TSAD built around relative contextual discrimination.

**Synthetic data for time series analysis** The paradigm of leveraging synthetic data for pre-training time series models has been extensively validated, particularly in the domain of forecasting, where it has led to significant performance gains across various benchmarks (Xie et al., 2025; Cai et al., 2024; Dooley et al., 2023; Hoo et al., 2025; Taga et al., 2025; Bhethanabhotla et al., 2024; Moroshan et al., 2025). Following this trend, several foundation models for general time series analysis have been successfully trained on a hybrid corpus of real-world and synthetic data to enhance their generalization capabilities (Ansari et al., 2024; Das et al., 2024; Shi et al., 2024; Goswami et al., 2024). In contrast, the application of purely synthetic data for TSAD remains substantially less explored. Existing efforts in this area have primarily focused on data augmentation, where artificial anomalies are injected into real-world time series to enrich the training set (Shentu et al., 2024; Darban et al., 2025). While effective, these methods are still fundamentally dependent on the availability and diversity of the underlying real data. Other studies have discussed the generation of entirely synthetic anomalous time series but have not applied this data to the end-to-end training of a dedicated TSAD model (Cai et al., 2024; Xie et al., 2024). To the best of our knowledge, this work is among the first to propose and implement a pre-training framework for a foundation model dedicated to TSAD that relies exclusively on a large-scale, fully synthetic dataset.

# B. Data Generation

Our synthetic data generation framework is designed as a curriculum to encourage broad generalization across diverse temporal patterns and causal couplings. The four-stage process first builds a complex, predictable "normal" context (Stages 1-2) and then introduces meaningful, context-violating anomalies (Stage 3). Finally, we generate causally aware ground-truth labels that teach the model to link an anomaly's root cause to its downstream effects (Stage 4), preparing it for zero-shot detection on unseen systems.

**Stage 1: Synthesizing the "Normal" Univariate Context** Every time series, whether it stands alone or is a component in a multivariate system, begins as a univariate baseline signal, $x_{\text{base}}(t)$. This signal is constructed as an additive composite of trend, seasonal, and noise components:

$$x_{\text{base}}(t) = T(t) + S(t) + \varepsilon(t), \quad t = 0, 1, \dots, n-1.$$

**Sequence length.** We sample $n$ from 100 to 10000 to match variable-length sequences.

**Trend $T(t)$.** We mix deterministic and stochastic trends: $T_{\text{det}}(t)$ and $T_{\text{stoc}}(t)$. Set $T(t) = (1 - \rho_T) T_{\text{det}}(t) + \rho_T T_{\text{stoc}}(t)$ with $\rho_T \in [0, 1]$. The detailed construction of the trend component is given in Appendix B.2.

**Seasonality $S(t)$.** A mixture of $K$ periodic atoms:

$$S(t) = \sum_{k=1}^{K} A_k \, w_k(2\pi f_k t + \varphi_k; \theta_k) \,,$$

where $w_k$ is selected from a library (sinusoid, square/triangle wave, wavelet atom), with amplitude $A_k > 0$, frequency $f_k$, phase $\varphi_k$, and shape parameters $\theta_k$ (e.g., duty cycle for square, mother/scale for wavelets). Frequencies may be harmonic $f_k = m_k f_0$ or incommensurate. The detailed construction of the seasonal component is given in Appendix B.3.

**Noise** $\varepsilon(t)$**.** Zero-mean noise with optional piecewise volatility:

$$\varepsilon(t) \sim \mathcal{N}\big(0,\, \sigma^2(t)\big), \qquad \sigma(t) = \sigma_0 \prod_{r=1}^{R} \big(1 + \upsilon_r \, \mathbf{1}_{[a_r, b_r)}(t)\big),$$

allowing volatility bursts over windows $[a_r, b_r)$. This captures heteroskedastic segments.

**Stage 2: Weaving the Multivariate Causal Context via a Discretized ODE System**  This stage creates a multivariate system with clear causal dependencies. We first conceptualize continuous-time causal dynamics (ODE) and then discretize the dynamics into an autoregressive with exogenous inputs (ARX) model using the Euler scheme.

1. **Generate a Causal Structure (DAG):** Sample a directed acyclic graph $G = (V, E)$ over $N$ nodes by first drawing an Erdős–Rényi graph and then orienting edges along a random topological order to avoid cycles. Edge presence is controlled by $p_{\mathrm{edge}} \in (0, 1)$. An edge $(j, i) \in E$ means series $j$ causally influences $i$.

2. **Define the Continuous-Time Causal Dynamics (ODE Model):** We model the evolution of the **causal influence** on each node, $z_i(t)$, as a first-order linear ODE. This equation describes how the influence on node $i$ changes over time based on its own internal state and forcing from its parent nodes $P(i)$:

$$\frac{dz_i(t)}{dt} = -\gamma_i z_i(t) + \sum_{j \in P(i)} \beta_{ij} x_j(t - \tau_{ij}) + C_i$$

   Here, $-\gamma_i z_i(t)$ represents an internal decay term, the sum represents the forcing from parent nodes $x_j$, and $C_i$ is a constant bias.

3. **Discretize to an ARX Model for Implementation:** With Euler and $\Delta t = 1$, the discrete ARX is

$$z_i[t] = a_i z_i[t-1] + \sum_{j \in P(i)} b_{ij} x_j[t - \ell_{ij}] + c_i, \quad |a_i| \le 0.8,$$

   where $a_i = 1 - \gamma_i$ encodes decay stability, delays $\ell_{ij} \in \{0, \ldots, \ell_{\max}\}$, and $b_{ij}$ are exogenous gains. We sample $a_i \sim \mathcal{U}[-0.8, 0.8]$, $\ell_{ij} \sim \mathcal{U}\{0, \ldots, \ell_{\max}\}$, and $b_{ij} \sim \mathcal{N}(0, \sigma_b^2)$ (variance scaled by in-degree).

4. **Mix the Signals:** The observed node mixes baseline with causal channel:

$$x_i[t] = (1 - \alpha_i)\, x_{\mathrm{base},i}(t) + \alpha_i\, z_i[t], \quad \alpha_i \in [0, 1].$$

   In vector form ($x[t], z[t] \in \mathbb{R}^N$),

$$z[t] = A\, z[t-1] + \sum_{\delta=0}^{\ell_{\max}} B^{(\delta)} x[t-\delta] + c, \quad x[t] = (I - A')\, x_{\mathrm{base}}(t) + A'\, z[t],$$

   where $A = \mathrm{diag}(a_i)$, $A' = \mathrm{diag}(\alpha_i)$, and $B^{(\delta)}$ encodes lagged influences.

This process, performed sequentially according to the topological sort, creates a system where a change in a parent node will naturally propagate to its child nodes after a specified delay.

**Stage 3: Principled Anomaly Injection: Simulating Causal and Contextual Complexity**  Our primary contribution in data generation lies in a synthetic framework designed to address critical gaps in existing anomaly detection benchmarks, namely the lack of causal realism and contextual complexity. To achieve this, our methodology introduces a novel causal framework for generating anomalies based on two distinct, complementary philosophies:

- **Endogenous Anomaly (Internal System Failure):** This strategy simulates how a single root-cause anomaly propagates through an interconnected system. An anomaly is introduced into the baseline series of a parent node *before* causal mixing. Its effects are then passed to descendant nodes based on the ARX model, creating a realistic cascade of causally-linked anomalous events.

- **Exogenous Anomaly (External Shock):** This mirrors more conventional techniques, modeling an external event impacting an otherwise normal system. The normal multivariate system is fully generated first, and then a segment of a single series is overwritten with an anomalous template.

**Stage 4: Label Generation and Causal Propagation** We generate token-level binary labels. For exogenous injections, positive labels correspond to the intervention window. For endogenous injections, we label the root-cause window and extend it to descendant channels based on their causal lags, as implied by the DAG and ARX lags $\{\ell_{ij}\}$. To capture the duration of these propagated effects, we use a decay time scale derived from the linear dynamics in Stage 2. Specifically, for the AR recurrence, we approximate the decay time scale as $t_{1/2} \approx \ln(2)/(1 - |a_i|)$. We define the effective anomaly window for descendant nodes by extending the root-cause duration by a factor of this decay time scale (e.g., $3 \times t_{1/2}$), ensuring the label encompasses the period where the causal shock remains significant before decaying. While our generation process tracks channel-specific anomalies (root-cause vs. propagated effects), for this work, we aggregate these into temporal localization labels (marking when an anomaly occurs across any channel). This aligns with standard zero-shot anomaly detection benchmarks, which typically provide timestep-level but not channel-level ground truth. Sequence lengths, DAG sparsity, ARX coefficients, and signal regimes (trend, seasonality, noise) are sampled from configurable priors (Appendix B.1), generating rich, interpretable dynamics designed for zero-shot learning.

In addition to causal dynamics, we move beyond simple point anomalies by creating a rich taxonomy of contextual anomalies designed to violate specific, established patterns within the data. These include:

- **Local/Change Anomalies:** Additive templates, $\Delta(t)$, applied over a window, representing events like a sudden sensor glitch (**Spike**), a component failure shifting its baseline (**Level Shift**), or mechanical vibrations (**Shake**).

$$x'_{\text{anom}}(t) = x_{\text{norm}}(t) + \Delta(t) \quad \text{for } t \in [t_s, t_e]$$

- **Seasonal Anomalies:** The normal seasonal component $S(t)$ is replaced with an altered version $S'(t)$. These anomalies simulate systemic issues such as a machine operating at the wrong speed (**Frequency Change**) or process desynchronization (**Phase Shift**).

$$x'_{\text{anom}}(t) = T(t) + S'(t) + \varepsilon(t) \quad \text{for } t \in [t_s, t_e]$$

In contrast to methods that inject isolated anomalies into independent time series, our dual-pronged approach provides a more challenging and realistic benchmark. By generating complex scenarios with both contextual violations and causal propagation, our dataset enables a more robust evaluation of a model's ability to understand system-wide dynamics. Details for constructing the anomaly types are provided in Appendix B.4.

## B.1. Hyperparameters and Sampling Mappings

To make the generation process transparent yet non-programmatic, we group hyperparameters into dataset-level controls (Table 4), attribute families and their sampling weights (Table 5), and multivariate graph and causal dynamics (Table 6).

*Table 4.* Dataset-level generation controls and intended effects.

| Name | Typical range / rule | Applies to | Intended effect |
|------|---------------------|------------|-----------------|
| Number of samples | positive integer | whole dataset | Total number of generated sequences. |
| Sequence length | uniformly in $\{100, \ldots, 10000\}$ | per sample | Time horizon of each sequence. |
| Anomalous-sample ratio | real in $[0, 1]$ | per sample | Probability a sample contains anomalies. |
| Multivariate flag | $\{$True, False$\}$ | per sample | Whether to generate multi-channel series. |
| Number of features | fixed or uniformly in $\{2, \ldots, 50\}$ | multivariate only | Channel count when multivariate. |
| Random seed (optional) | user-provided integer | whole run | Reproducibility of the random draws. |

## B.2. Trend Types and Formal Definitions

We instantiate the trend component $T(t)$ using five archetypes aligned with the generator configuration. Let $t \in \{0, \ldots, n - 1\}$.

**Increase / Decrease / Constant.** We use an affine model

$$T(t) = k_0 + k_1 t,$$

with constraints: *increase* uses $k_1 > 0$, *decrease* uses $k_1 < 0$, and *constant* uses $k_1 = 0$ (constant baseline).

*Table 5.* Attribute categories, options, and relative probability.

| Category | Options | Probability |
|---|---|---|
| Seasonality type | no periodic fluctuation | 0.3 |
| | sine | 0.3 |
| | square | 0.05 |
| | triangle | 0.05 |
| | wavelet | 0.3 |
| Trend type | decrease | 0.2 |
| | increase | 0.2 |
| | constant | 0.2 |
| | multiple | 0.3 |
| | ARIMA | 0.1 |
| Frequency regime | high frequency | 0.5 |
| | low frequency | 0.5 |
| Noise level | almost no noise | 0.25 |
| | low | 0.25 |
| | moderate | 0.25 |
| | high | 0.25 |

*Table 6.* Multivariate and causal-dynamics hyperparameters and intended roles.

| Name | Typical range / rule | Applies to | Intended role |
|---|---|---|---|
| Number of nodes | Uniformly in 1 to 50 | system graph | Dimensionality of the multivariate system. |
| Number of edges | roughly linear in nodes (acyclic) | system graph | Connectivity while preserving acyclicity. |
| Parent lags | nonnegative integers up to a small fraction of length | per edge | Causal delays between channels. |
| ARX autoregression $a_i$ | real in $[-0.8, 0.8]$ | per node | Decay/stability of latent causal influence. |
| ARX exogenous gains $b_{ij}$ | zero-mean with finite variance | per edge | Strength of parent-to-child coupling. |
| ARX bias $c_i$ | bounded real interval | per node | Baseline offset in causal channel. |
| Mixing weight $\alpha_i$ | real in $[0, 1]$ | per node | Balance between baseline and causal channel. |

**Multiple (piecewise trend).** To capture mixed regimes, we allow $P$ change-points at integer knots $0 < \tau_1 < \cdots < \tau_P < n$ and define a piecewise-linear trend with slope updates:

$$T(t) = k_0 + k_1 t + \sum_{p=1}^{P} \delta_p \left[ t \geq \tau_p \right] (t - \tau_p)_+,$$

where $(u)_+ = \max\{u, 0\}$ and $\delta_p$ adjusts slope after each $\tau_p$. This matches the additive hinge construction used in Stage 1.

**ARIMA (stochastic trend).** For stochastic trends we use a differenced ARMA process. Let $\nabla^d$ be the $d$-th difference operator. A general ARIMA$(p, d, q)$ satisfies

$$\phi(B) \, \nabla^d T(t) = \theta(B) \, \varepsilon_T(t), \qquad \varepsilon_T(t) \sim \mathcal{N}(0, \sigma_T^2),$$

with backshift $B$, AR polynomial $\phi(B) = 1 - \sum_{i=1}^{p} \phi_i B^i$, and MA polynomial $\theta(B) = 1 + \sum_{j=1}^{q} \theta_j B^j$. In practice we use small orders (e.g., $p, q \leq 2$) and modest variance.

### B.3. Seasonality Types and Formal Definitions

We support five seasonal archetypes. Let the base period be $P > 0$ (in samples), frequency $f = 1/P$, amplitude $A > 0$, and phase $\varphi \in [0, 2\pi)$.

**No periodic fluctuation.** No seasonality is present:

$$S(t) = 0.$$

**Sin periodic fluctuation.** A sinusoidal component:

$$S(t) = A \sin(2\pi f t + \varphi).$$

**Square periodic fluctuation.** A rectangular wave with duty cycle $d \in (0,1)$:

$$S(t) = A \, \mathrm{sgn}\big(\sin(2\pi f t + \varphi)\big),$$

or equivalently by its Fourier series $S(t) = \frac{4A}{\pi} \sum_{m \, \mathrm{odd}} m^{-1} \sin(2\pi m f t + \varphi)$; duty-cycle variants modulate on/off durations per period.

**Triangle periodic fluctuation.** A symmetric triangle wave of period $P$:

$$S(t) = \frac{8A}{\pi^2} \sum_{m \, \mathrm{odd}} \frac{(-1)^{(m-1)/2}}{m^2} \sin(2\pi m f t + \varphi),$$

equivalently realizable as a piecewise-linear ramp that rises and falls within each period.

**Wavelet periodic fluctuation.** A sum of localized atoms (e.g., `db`, `sym`, `coif`, `haar`, `bior`, `dmey` families):

$$S(t) = \sum_{\ell=1}^{L} A_\ell \, \psi_\ell\left(\frac{t - \tau_\ell}{s_\ell}\right), \qquad A_\ell \in \mathbb{R}, \; s_\ell > 0, \; \tau_\ell \in \mathbb{R}.$$

### B.4. A Taxonomy of Synthetic Anomaly Archetypes

Following the high-level description, we now provide complete, self-contained definitions and formulas for each anomaly type supported by our generator. We divide them into two families: Local/Change Anomalies (abrupt, windowed changes) and Seasonal Anomalies (contextual violations of periodic structure). We avoid shorthand; every symbol is explicitly defined. A comprehensive summary of all anomaly archetypes with intents and formal definitions is provided in Table 7.

**Notation and primitives.** We use discrete time index $t \in \{0, 1, \ldots, n-1\}$. The normal (anomaly-free) series is denoted by $x_{\mathrm{norm}}(t)$. An anomaly acts on a closed-open window $[t_s, t_e)$ with integers $0 \le t_s < t_e \le n$. The indicator function is

$$\mathbf{1}_{[t_s, t_e)}(t) = \begin{cases} 1, & t_s \le t < t_e, \\ 0, & \text{otherwise.} \end{cases}$$

The Heaviside step function at integer times is $H(t; t_0) = \mathbf{1}_{[t_0, \infty)}(t)$. For any scalar $u$, the logistic sigmoid is $\sigma(u) = \frac{1}{1+e^{-u}}$. For a rectangular window we use $h(t; t_s, t_e) = \mathbf{1}_{[t_s, t_e)}(t)$. All amplitudes $A, B, \{A_m\}$ are real scalars; all durations and lags (e.g., $w, \tau_r, \tau_f, d$) are positive integers unless otherwise stated.

#### B.4.1. LOCAL/CHANGE ANOMALIES

All local anomalies are applied additively within a window using a template $\Delta(t)$:

$$x'_{\mathrm{anom}}(t) = x_{\mathrm{norm}}(t) + \Delta(t) \, \mathbf{1}_{[t_s, t_e)}(t), \quad t = 0, 1, \ldots, n-1.$$

**Spikes and bursts**

- **Upward spike** (a single sharp positive glitch). Parameters: center time $t_0 \in [t_s, t_e)$, half-width $w \in \mathbb{N}^+$, amplitude $A > 0$. Define

$$\Delta(t) = A \, \max\left(1 - \frac{|t - t_0|}{w}, 0\right), \quad t \in \mathbb{Z}.$$

- **Downward spike** (a single sharp negative glitch). Same parameters as above with negative amplitude:

$$\Delta(t) = -A \, \max\left(1 - \frac{|t - t_0|}{w}, 0\right).$$

- **Continuous upward spikes** (a burst of multiple positive spikes). Parameters: spike count $M \in \mathbb{N}^+$, centers spaced by stride $d \in \mathbb{N}^+$, per-spike amplitudes $A_m > 0$, per-spike widths $w_m \in \mathbb{N}^+$. Define

$$\Delta(t) = \sum_{m=0}^{M-1} A_m \, \max\left(1 - \frac{|t - (t_0 + md)|}{w_m}, 0\right).$$

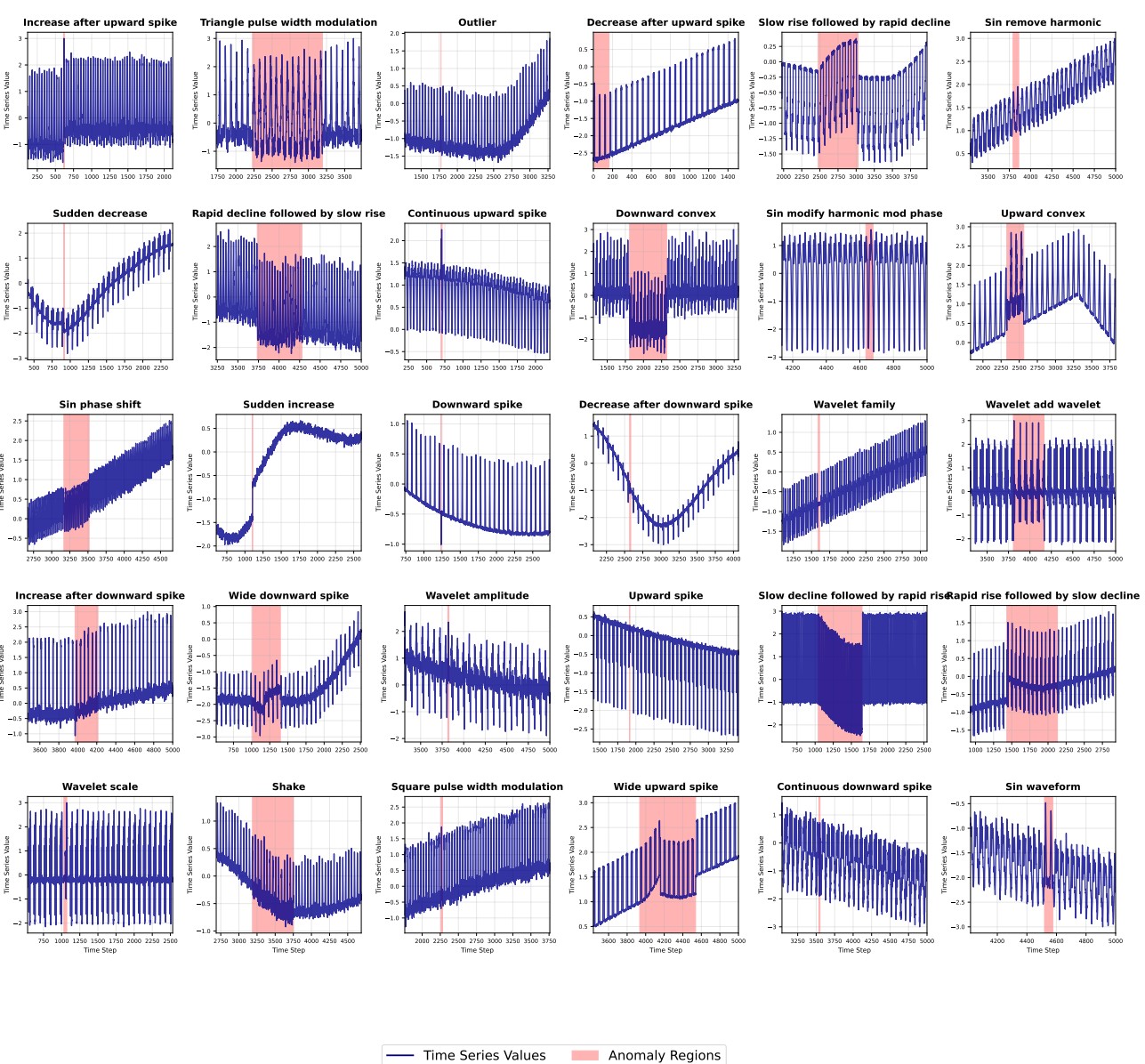

*Figure 9.* Illustration of synthetic anomaly archetypes.

- **Continuous downward spikes** (a burst of negative spikes). Use

$$\Delta(t) = \sum_{m=0}^{M-1} (-A_m) \max\left(1 - \frac{|t - (t_0 + md)|}{w_m}, 0\right).$$

- **Wide upward spike** (longer rise, short plateau, and fall). Parameters: rise length $\tau_r \in \mathbb{N}^+$, fall length $\tau_f \in \mathbb{N}^+$, amplitude $A > 0$. Define

$$\Delta(t) = \begin{cases} A \frac{t - t_s}{\tau_r}, & t \in [t_s, t_s + \tau_r), \\ A, & t \in [t_s + \tau_r, t_e - \tau_f), \\ A\left(1 - \frac{t - (t_e - \tau_f)}{\tau_f}\right), & t \in [t_e - \tau_f, t_e), \\ 0, & \text{else.} \end{cases}$$

- **Wide downward spike** (longer fall then recovery). Use the same piecewise definition with amplitude $-A$.

- **Outlier** (single-point shock). Parameters: position $t_0 \in [t_s, t_e)$, amplitude $A \in \mathbb{R}$. Define the Kronecker impulse

$$\Delta(t) = \begin{cases} A, & t = t_0, \\ 0, & t \neq t_0. \end{cases}$$

## Level shifts and plateaus

- **Sudden increase** (permanent level up-shift). Parameters: slope parameter $\kappa > 0$, amplitude $A > 0$, change time $t_0 \in [t_s, t_e)$. Define

$$\Delta(t) = A\,\sigma\big(\kappa(t - t_0)\big), \quad \sigma(u) = \frac{1}{1 + e^{-u}}.$$

- **Sudden decrease** (permanent level down-shift). Use

$$\Delta(t) = -A\,\sigma\big(\kappa(t - t_0)\big).$$

- **Convex/Concave plateau** (temporary level change using a raised-cosine window). Parameters: window $[t_s, t_e)$, amplitude $A > 0$, sign $s \in \{+1, -1\}$ (convex when $s = +1$, concave when $s = -1$). Define

$$\Delta(t) = s\,A\,\tfrac{1}{2}\left(1 - \cos\frac{\pi(t - t_s)}{t_e - t_s}\right)\mathbf{1}_{[t_s, t_e)}(t).$$

## Asymmetric transients

- **Rapid rise followed by slow decline**. Parameters: rise time constant $\tau_r \in \mathbb{N}^+$, fall time constant $\tau_f \in \mathbb{N}^+$, peak time $t_p \in [t_s, t_e]$, amplitude $A > 0$. Define

$$\Delta(t) = A\left(1 - e^{-\frac{t - t_s}{\tau_r}}\right)\mathbf{1}_{[t_s, t_p)}(t) + A\,e^{-\frac{t - t_p}{\tau_f}}\,\mathbf{1}_{[t_p, t_e)}(t), \quad \tau_r \ll \tau_f.$$

- **Slow rise followed by rapid decline**. Use the same definition with $\tau_r \gg \tau_f$.

- **Rapid decline followed by slow rise**. Use the above definition with amplitude $-A$ and swap rise/fall roles accordingly.

- **Slow decline followed by rapid rise**. Use the rapid rise–slow decline formula with amplitude $-A$ and $\tau_r \gg \tau_f$.

## Spike + level interaction

- **Decrease after upward spike**. Parameters: spike template $\Delta_{\text{spk}}(t)$ (e.g., the triangular spike above), post-spike shift magnitude $B > 0$, shift start $t_1 \geq t_s$. Define

$$\Delta(t) = \Delta_{\text{spk}}(t) - B\,H(t; t_1), \quad H(t; t_1) = \mathbf{1}_{[t_1, \infty)}(t).$$

- **Increase after downward spike**. Using a negative spike template $\Delta_{\mathrm{nspk}}(t)$ and a positive shift $B > 0$:

$$\Delta(t) = \Delta_{\mathrm{nspk}}(t) + B\,H(t; t_1).$$

- **Increase after upward spike**. Use

$$\Delta(t) = \Delta_{\mathrm{spk}}(t) + B\,H(t; t_1).$$

- **Decrease after downward spike**. Use

$$\Delta(t) = \Delta_{\mathrm{nspk}}(t) - B\,H(t; t_1).$$

**High-frequency burst**

- **Shake** (localized high-frequency vibration). Parameters: carrier frequency $f_h \in \mathbb{R}_{>0}$ in cycles per sample, phase $\varphi \in [0, 2\pi)$, amplitude $A > 0$. With a rectangular window $h(t; t_s, t_e)$,

$$\Delta(t) = A\,h(t; t_s, t_e)\,\sin(2\pi f_h t + \varphi).$$

B.4.2. SEASONAL ANOMALIES

Seasonal anomalies are implemented by replacing the seasonal component $S(t)$ with a modified component $S'(t)$ strictly within the window $[t_s, t_e)$, while leaving the rest unchanged:

$$x'_{\mathrm{anom}}(t) = \begin{cases} T(t) + S'(t) + \varepsilon(t), & t_s \le t < t_e, \\ T(t) + S(t) + \varepsilon(t), & \text{otherwise.} \end{cases}$$

**Universal transforms (any waveform)**

- **Waveform inversion** (polarity flip). For $t \in [t_s, t_e)$, set $S'(t) = -S(t)$.

- **Amplitude scaling**. For $t \in [t_s, t_e)$, set $S'(t) = r\,S(t)$ with scaling factor $r > 0$.

- **Frequency change**. Let the original period be $P > 0$ and frequency $f = 1/P$. Choose a period multiplier $\rho > 0$ and set the new period $P' = \rho P$ and frequency $f' = 1/P'$. For a sinusoid $S(t) = A\sin(2\pi f t + \varphi)$, define $S'(t) = A\sin(2\pi f' t + \varphi)$ for $t \in [t_s, t_e)$.

- **Noise injection**. For $t \in [t_s, t_e)$, set $S'(t) = S(t) + \eta(t)$ with i.i.d. Gaussian noise $\eta(t) \sim \mathcal{N}(0, \sigma_S^2)$.

- **Waveform change from sine to square/triangle**. If $S(t)$ is sinusoidal with amplitude $A$ and period $P$, replace it by a square/triangle waveform $w_{\mathrm{new}}(t; A, P, \theta)$ of the same amplitude and period (with shape parameters $\theta$) for $t \in [t_s, t_e)$.

**Sinusoidal-specific (harmonics and phase)** Let a sinusoidal seasonal component be represented by a finite harmonic sum

$$S(t) = \sum_{n=1}^{N} A_n \sin\big(2\pi n f\, t + \varphi_n\big),$$

and optionally include amplitude modulation for each harmonic via

$$A_n(t) = \frac{A_n}{n}\,(1 + d_n \sin(\omega_m t + \psi_n))\,, \quad d_n \in [0, 1),\ \omega_m > 0,\ \psi_n \in [0, 2\pi).$$

- **Phase shift**. For $t \in [t_s, t_e)$, update all phases by a fixed shift $\Delta\varphi \in (0, 2\pi)$: $\varphi_n \mapsto \varphi_n + \Delta\varphi$.

- **Add harmonic**. For $t \in [t_s, t_e)$, add a new harmonic of order $m \in \mathbb{N}^+$: $S'(t) = S(t) + A_h \sin(2\pi m f\, t + \varphi_h)$ with amplitude $A_h > 0$ and phase $\varphi_h \in [0, 2\pi)$.

- **Remove harmonic**. For $t \in [t_s, t_e)$, remove one existing harmonic index $n^* \in \{1, \ldots, N\}$ from the sum, keeping the others unchanged.

- **Modify harmonic phase**. For $t \in [t_s, t_e)$, choose one index $n^*$ and set $\varphi_{n^*} \mapsto \varphi'_{n^*}$ such that $|\varphi'_{n^*} - \varphi_{n^*}| \geq \delta$ for a given threshold $\delta > 0$.

- **Modify amplitude-modulation depth**. For $t \in [t_s, t_e)$, choose index $n^*$ and set $d_{n^*} \mapsto d'_{n^*}$ within $[0, 1)$.

- **Modify modulation frequency**. For $t \in [t_s, t_e)$, update $\omega_m \mapsto \omega'_m$ with $\omega'_m > 0$.

- **Modify modulation phase**. For $t \in [t_s, t_e)$, choose index $n^*$ and set $\psi_{n^*} \mapsto \psi'_{n^*}$.

**Square/Triangle-specific (pulse geometry)** For a square or triangle waveform with period $P > 0$, define the duty cycle $d \in (0, 1)$ as the fraction of each period the waveform is at its high level, and define a cycle start parameter $\delta \in [0, 1)$ that shifts the waveform phase relative to each period.

- **Pulse shift**. For $t \in [t_s, t_e)$, apply a phase shift $\Delta \in (-1, 1)$ cycles: $\delta \mapsto (\delta + \Delta) \bmod 1$.

- **Pulse width modulation**. For $t \in [t_s, t_e)$, scale the duty cycle by a factor $\lambda > 0$: $d \mapsto d' = \min\{1, \max\{0, \lambda d\}\}$.

**Wavelet-based (atoms and families)** Let a wavelet-based seasonal component be represented as a finite sum of atoms

$$S(t) = \sum_{\ell=1}^{L} A_\ell \, \psi_\ell \left( \frac{t - \tau_\ell}{s_\ell} \right), \quad A_\ell \in \mathbb{R}, \ s_\ell > 0, \ \tau_\ell \in \mathbb{R}.$$

- **Family change**. For $t \in [t_s, t_e)$, replace each mother wavelet $\psi_\ell$ taken from a family $\mathcal{F}$ by a corresponding wavelet $\tilde{\psi}_\ell$ from a different family $\tilde{\mathcal{F}}$ while keeping $A_\ell, s_\ell, \tau_\ell$ unchanged.

- **Scale change**. For $t \in [t_s, t_e)$, update scales by a factor $\lambda > 0$: $s_\ell \mapsto s'_\ell = \lambda s_\ell$ for selected indices.

- **Shift change**. For $t \in [t_s, t_e)$, update translations by $\tau_\ell \mapsto \tau'_\ell = \tau_\ell + \Delta_{\tau, \ell}$.

- **Amplitude change**. For $t \in [t_s, t_e)$, rescale coefficients by $A_\ell \mapsto r_\ell A_\ell$ with $r_\ell > 0$.

- **Add wavelet**. For $t \in [t_s, t_e)$, append a new atom $A_+ \psi_+ \left( \frac{t - \tau_+}{s_+} \right)$ with specified parameters.

- **Remove wavelet**. For $t \in [t_s, t_e)$, remove one existing atom index from the sum.

## C. Mathematical Formulation of Relative Context Discrepancy

### C.1. Definition of RCD

Some widely used anomaly-detection pipelines score timesteps largely through local deviations, for example by measuring deviation from expected values or by assessing reconstruction errors. *Relative Context Discrepancy (RCD)* frames the problem differently. Rather than focusing on the individual value of a token, it evaluates whether the *relationships* a token forms with its surrounding context remain consistent with normal relational patterns. Under this view, an anomaly emerges as a disruption in these relational structures, and RCD therefore emphasizes relational inconsistency rather than absolute deviation.

### C.2. Mathematical Formulation of RCD

Consider a time-series sequence

$$X_1, X_2, \ldots, X_T,$$

where each $X_t$ denotes the observation at time $t$. RCD does not describe $X_t$ by its value alone, but by how it interacts with the other tokens in the sequence.

*Table 7.* Summary of anomaly archetypes, intents, and formal definitions.

| Family | Type | Intent (one sentence) | Formal definition (within $[t_s, t_e)$) |
|---|---|---|---|
| Local | Upward spike | A single sharp positive glitch | $\Delta(t) = A \max(1 - |t - t_0|/w, 0)$ |
| | Downward spike | A single sharp negative glitch | $\Delta(t) = -A \max(1 - |t - t_0|/w, 0)$ |
| | Continuous upward spikes | Rapid consecutive positive spikes | $\sum_{m=0}^{M-1} A_m \max(1 - |t - (t_0 + md)|/w_m, 0)$ |
| | Continuous downward spikes | Rapid consecutive negative spikes | $\sum_{m=0}^{M-1} (-A_m) \max(1 - |t - (t_0 + md)|/w_m, 0)$ |
| | Wide upward spike | Longer rise/plateau/fall pulse | piecewise linear rise/hold/fall with amplitude $A$ |
| | Wide downward spike | Longer fall/plateau/rise pulse | piecewise linear with amplitude $-A$ |
| | Outlier | Single-point shock | $\Delta(t) = A$ if $t = t_0$, else 0 |
| | Sudden increase | Permanent upward level shift | $\Delta(t) = A \sigma(\kappa(t - t_0))$ |
| | Sudden decrease | Permanent downward level shift | $\Delta(t) = -A \sigma(\kappa(t - t_0))$ |
| | Convex/Concave plateau | Temporary level change | $sA\frac{1}{2}\big(1 - \cos(\pi(t - t_s)/(t_e - t_s))\big)$ |
| | Rapid rise → slow decline | Fast excitation then slow decay | $A(1 - e^{-(t-t_s)/\tau_r})\mathbf{1}_{[t_s, t_p]} + Ae^{-(t-t_p)/\tau_f}\mathbf{1}_{[t_p, t_e)}$ |
| | Slow rise → rapid decline | Slow excitation then fast decay | same as above with $\tau_r \gg \tau_f$ |
| | Rapid decline → slow rise | Fast drop then slow recovery | negative-amplitude version of above |
| | Slow decline → rapid rise | Slow drop then fast recovery | negative-amplitude version of above |
| | Decrease after upward spike | Spike followed by downward shift | $\Delta_{\text{spk}}(t) - B\,H(t; t_1)$ |
| | Increase after downward spike | Negative spike then upward shift | $\Delta_{\text{nspk}}(t) + B\,H(t; t_1)$ |
| | Increase after upward spike | Spike followed by upward shift | $\Delta_{\text{spk}}(t) + B\,H(t; t_1)$ |
| | Decrease after downward spike | Negative spike then downward shift | $\Delta_{\text{nspk}}(t) - B\,H(t; t_1)$ |
| | Shake | Localized high-frequency vibration | $A \sin(2\pi f_h t + \varphi)$ within $[t_s, t_e)$ |
| Seasonal | Waveform inversion | Polarity flip of seasonal component | $S'(t) = -S(t)$ |
| | Amplitude scaling | Change seasonal magnitude | $S'(t) = r\,S(t)$ |
| | Frequency change | Alter base period/frequency | $S'(t)$ with $P' = \rho P$ (e.g., $f' = 1/P'$) |
| | Noise injection | Add Gaussian noise to seasonality | $S'(t) = S(t) + \eta(t),\ \eta \sim \mathcal{N}(0, \sigma_S^2)$ |
| | Waveform change (sin→square/triangle) | Replace waveform family | $S'(t) = w_{\text{new}}(t; A, P, \theta)$ |
| | Phase shift (sinusoid) | Global phase offset | $\varphi_n \mapsto \varphi_n + \Delta\varphi$ |
| | Add harmonic (sinusoid) | Introduce new harmonic | $S'(t) = S(t) + A_h \sin(2\pi m f t + \varphi_h)$ |
| | Remove harmonic (sinusoid) | Remove one harmonic | drop index $n^*$ term in $\sum_n$ |
| | Modify harmonic phase | Change one harmonic phase | $\varphi_{n^*} \mapsto \varphi'_{n^*}$ |
| | Modify amp-mod depth | Change amplitude modulation depth | $d_{n^*} \mapsto d'_{n^*}$ |
| | Modify modulation frequency | Change modulation frequency | $\omega_m \mapsto \omega'_m$ |
| | Modify modulation phase | Change modulation phase | $\psi_{n^*} \mapsto \psi'_{n^*}$ |
| | Pulse shift (square/triangle) | Shift cycle start | $\delta \mapsto (\delta + \Delta) \bmod 1$ |
| | Pulse width modulation | Change duty cycle | $d \mapsto d' = \min\{1, \max\{0, \lambda d\}\}$ |
| | Wavelet family change | Swap mother wavelet family | $\psi_\ell \in \mathcal{F} \to \tilde{\psi}_\ell \in \tilde{\mathcal{F}}$ |
| | Wavelet scale change | Rescale atom width | $s_\ell \mapsto s'_\ell = \lambda s_\ell$ |
| | Wavelet shift change | Shift atom translation | $\tau_\ell \mapsto \tau'_\ell$ |
| | Wavelet amplitude change | Rescale atom amplitude | $A_\ell \mapsto r_\ell A_\ell$ |
| | Add wavelet atom | Add new atom | append $(A_+, s_+, \tau_+, \psi_+)$ |
| | Remove wavelet atom | Remove existing atom | delete one summand index |

To formalize this, introduce two functions. The first, $f(X_j, X_t)$, is a pairwise interaction function representing how token $X_j$ contributes to the interpretation of token $X_t$. This function captures the *relative* aspect of RCD by explicitly modeling token-to-token relationships:

$$f(X_j, X_t).$$

Anomalies, however, typically arise not from a single unusual interaction, but from an *unusual pattern* of interactions. This motivates an aggregation function $g$ that synthesizes all pairwise contributions into a contextual embedding for token $t$:

$$\text{embedding}_{X_t} = g\big(f(X_1, X_t), f(X_2, X_t), \ldots, f(X_T, X_t)\big).$$

This embedding summarizes the relational profile that determines how token $t$ fits into its broader temporal context. Under this formulation, an anomaly corresponds to an embedding that deviates from those produced under normal relational patterns.

The combination of $f$ and $g$ therefore operationalizes the central RCD principle: *the meaning of a token arises from its relationships with the rest of the sequence*.

## C.3. Instantiation in TimeRCD via Self-Attention

The RCD formulation aligns naturally with the mechanism of self-attention. In a Transformer, each token $X_t$ produces a query vector

$$q_t = X_t W^Q,$$

while each token $X_j$ generates

$$k_j = X_j W^K, \qquad v_j = X_j W^V.$$

The pairwise interaction between tokens $t$ and $j$ is expressed through the scaled dot-product score

$$s_{tj} = \frac{q_t \cdot k_j}{\sqrt{d_k}}.$$

The corresponding unnormalized contribution can be written as

$$f(X_j, X_t) = v_j \, \exp(s_{tj}),$$

which directly instantiates the pairwise interaction function $f$.

Self-attention aggregates these contributions using softmax normalization, which serves as a concrete realization of the aggregation function $g$:

$$\text{embedding}_{X_t} = \frac{\sum_j v_j \exp\left(\frac{q_t \cdot k_j}{\sqrt{d_k}}\right)}{\sum_j \exp\left(\frac{q_t \cdot k_j}{\sqrt{d_k}}\right)}.$$

Within the RCD perspective, discrepancy corresponds to deviations in the attention pattern of token $X_t$ relative to the relational patterns learned from pre-training data. When a token is anomalous, its query–key interactions produce an atypical similarity distribution $s_{tj}$, leading to a context embedding that lies outside the manifold of embeddings observed during normal behavior. The anomaly head is then trained to map such deviations in the embedding to a high anomalous probability, thereby converting relational irregularity directly into an anomaly score.

## C.4. Anomaly Head, Reconstruction Head, and Training Objective

TimeRCD employs two heads during training: an anomaly head and a reconstruction head. The anomaly head applies a lightweight MLP to each contextual embedding and outputs logits for the normal and anomalous classes:

$$\text{logits}_{x_t} = \text{AnomalyHead}(\text{embedding}_{X_t}),$$

and the anomaly score is defined as the softmax probability of the anomalous class:

$$\text{AnomalyScore}_{x_t} = \text{softmax}(\text{logits}_{x_t})[1].$$

The reconstruction head is a 3-layer MLP that predicts a scalar reconstruction for each timestep:

$$\widehat{x}_t = \text{ReconstructionHead}(\text{embedding}_{X_t}).$$

The total loss combines a cross-entropy loss for anomaly prediction and an MSE loss for reconstruction:

$$L_{\text{total}} = \lambda_{\text{anomaly}} \cdot \text{CrossEntropy}(\text{logits}_{x_t}, y_t) + \lambda_{\text{recon}} \cdot \text{MSE}(\widehat{x}_t, x_t),$$

where $y_t$ and $x_t$ are the label and value at timestamp $t$, respectively. In all experiments, both weighting coefficients are set to $\lambda_{\text{anomaly}} = \lambda_{\text{recon}} = 1$.

# D. Details of experiments

## D.1. Benchmark Datasets

Our evaluation uses a selection of datasets from one primary source, the TSB-AD benchmark by (Liu & Paparrizos, 2024). Our dataset selection prioritizes evaluation quality and avoids redundancy. A substantial fraction of the TSB-AD benchmark's "univariate" section consists of decomposed multivariate datasets (where dimensions of multivariate series are treated as separate univariate ones), which can introduce bias and inflate performance metrics. To ensure a rigorous and unbiased evaluation, we deliberately filtered out these decomposed series, focusing on **genuinely univariate** datasets for the univariate track and selecting five standard multivariate datasets for the multivariate track (testing them in their original form). This selection covers diverse domains (e.g., medical, industrial, server metrics, finance, weather) while preventing "double counting" and providing a clearer picture of model capabilities.

- **Univariate Datasets**: We utilize a diverse collection of univariate datasets including IOPS, MGAB, NAB, NEK, Power, SED, Stock, TODS, UCR, WSD, and YAHOO.

- **Multivariate Datasets**: For multivariate anomaly detection, we use the MSL, PSM, SMAP, SMD, and SWaT datasets. These are sourced from industrial control systems and spacecraft telemetry, presenting complex, multi-dimensional dependencies.

The specific characteristics of these datasets, including their domain, number of time series (TS), average length, and anomaly ratio, are summarized in Table 8 and Table 9.

*Table 8.* Univariate Datasets

| Name | Domain | #TS | Avg Length | AR (%) |
|------|--------|-----|-----------|--------|
| UCR | Misc. | 228 | 67818.7 | 0.6 |
| NAB | Web | 28 | 5099.7 | 10.6 |
| YAHOO | Web | 259 | 1560.2 | 0.6 |
| IOPS | Operations | 17 | 72792.3 | 1.3 |
| MGAB | Sensor | 9 | 97777.8 | 0.2 |
| SED | Energy | 3 | 23332.3 | 4.1 |
| Stock | Finance | 20 | 15000 | 9.4 |
| TODS | Traffic | 15 | 5000.0 | 6.3 |
| NEK | Weather | 9 | 1073.0 | 8.0 |
| Power | Power Grid | 1 | 35040.0 | 8.5 |
| WSD | Weather | 111 | 17444.5 | 0.6 |

*Table 9.* Multivariate Datasets

| Name | Domain | #TS | Avg Length | AR (%) |
|------|--------|-----|-----------|--------|
| MSL | Space | 16 | 3119.4 | 5.1 |
| PSM | Sensor | 1 | 217624.0 | 11.2 |
| SMAP | Space | 27 | 7855.9 | 2.9 |
| SMD | Server | 22 | 25466.4 | 3.8 |
| SWaT | ICS | 2 | 207457.5 | 12.7 |

## D.2. Baselines

Our study compares a range of baseline models, categorized into zero-shot and full-shot detectors, following the methodology from TSB-AD benchmark (Liu & Paparrizos, 2024) and the DADA (Shentu et al., 2024).

**Zero-shot models.** These models are pre-trained on large-scale datasets and can be applied directly to new time series without target-domain training.

- **DADA**: This model, from the paper by (Shentu et al., 2024), is a pre-trained general anomaly detector. We set the window size to 100.

- **MOMENT**: A foundation model that utilizes a patch-based pre-training strategy to learn rich representations from diverse time series data (Goswami et al., 2024). We use a window size of 64.

- **TimesFM**: A decoder-only transformer model from Google trained on a large time series corpus using a patching strategy, enabling strong zero-shot generalization (Das et al., 2024). We set the window size to 96.

- **Chronos**: A generative model that frames time series analysis as a language modeling task, using a transformer-based architecture to learn and predict time series values (Ansari et al., 2024). Its window size is 100.

- **Time MOE**: A decoder-only transformer model with a sparse Mixture-of-Experts (MoE) architecture. It is pre-trained on a large time series corpus for forecasting and multi-task learning (Shi et al., 2024). We set the window size to 96.

- **MovingVar.**: A simple statistical detector that scores each timestamp by the variance inside a moving window. We include it as a lightweight, non-neural baseline under the same evaluation protocol to test whether local variance shifts can account for the observed performance.

**Full-shot models.** These models require training on the target dataset.

- **TranAD**: A transformer-based model that uses a reconstructive approach to detect anomalies by comparing original and reconstructed time series (Tuli et al., 2022). It is configured with a window size of 10.

- **USAD**: An autoencoder-based model that employs an adversarial training mechanism to enhance its reconstruction capability and anomaly detection (Audibert et al., 2020). We set the window size to 100.

- **OmniAnomaly**: A deep learning model that uses a Variational Autoencoder (VAE) with a Gated Recurrent Unit (GRU) to learn normal patterns and detect deviations (Su et al., 2019). The model is configured with a window size of 100.

- **Sub-PCA**: A subspace-based method that detects anomalies by identifying deviations in principal component subspaces (Liu & Paparrizos, 2024). The model decomposes multivariate time series into subspaces and flags anomalies based on reconstruction errors in these subspaces.

- **DCdetector**: A deep learning approach that combines dual attention mechanisms and contrastive learning for time series anomaly detection (Yang et al., 2023). The model uses both temporal and channel-wise attention to capture complex dependencies.

- **TFMAE**: A transformer-based masked autoencoder that learns representations by reconstructing masked time series segments (Fang et al., 2024). The model leverages self-supervised pre-training to learn robust temporal patterns for anomaly detection.

- **LOF**: A traditional statistical method that measures the local deviation of a data point from its neighbors, identifying anomalies with lower local density (Breunig et al., 2000). We set the number of neighbors to $n = 50$ for both univariate and multivariate settings and use the Euclidean metric.

- **IForest**: An ensemble of Isolation Trees that isolates anomalies based on the number of random partitions required to separate them from the rest of the data (Liu et al., 2008). For univariate datasets, we set n estimators to 200, and for multivariate, we set n estimators to 25 and max features to 0.8.

### D.3. Pre-training Overlap with Evaluation Benchmarks

We exclude certain baseline results (specifically from DADA and MOMENT) because their pre-training corpora heavily overlap with our zero-shot test benchmarks, violating the zero-shot evaluation setting.

- DADA: Its pre-training data includes IOPS, MGAB, and YAHOO (Shentu et al., 2024).

- MOMENT: Uses the "Time Series Pile," which contains IOPS, MGAB, NAB, YAHOO, MSL, SMAP, and SMD (Goswami et al., 2024).

## D.4. Evaluation Metric Calculations

Our performance evaluation is conducted using four metrics: Affiliation F1 ($F1_A$), Temporal F1 ($F1_T$), Standard F1 ($F1$), and VUS-PR. We adopt a selection of metrics from the TSB-AD benchmark (Liu & Paparrizos, 2024) for Affiliation F1 ($F1_A$), Standard F1 ($F1$), and VUS-PR. For Temporal F1, we follow the implementation of (Sarfraz et al., 2024).

- **Standard-F1** The Standard F1-score is a widely used metric that provides a harmonic mean of precision and recall. It is calculated using point-wise True Positives (TP), False Positives (FP), and False Negatives (FN).

- **F1-T** Temporal F1, as described by (Sarfraz et al., 2024), is a range-based metric that evaluates anomaly detection performance by considering the temporal context of anomalies. It is a variant of the F1-score that addresses common issues in time series evaluation, such as overlapping predictions and temporal proximity.

- **Affiliation-F** ($F1_A$) Affiliation F1 is a distance-based metric that measures the "affiliation" or proximity between the ground truth and predicted anomaly points. It is designed to be less sensitive to minor temporal shifts in the predicted anomalies. The score is calculated by finding the optimal one-to-one mapping between the ground truth and detected anomalies and then computing the F1-score based on these affiliations (Huet et al., 2022).

- **VUS-PR** (Volume Under Surface - Precision/Recall) is a threshold-independent, parameter-free metric for time series anomaly detection. Unlike point-wise metrics, VUS-PR is robust to time lags and measures the area under a 3D surface plot of precision, recall, and a buffer parameter (Paparrizos et al., 2022). It addresses the limitations of standard F1 scores by creating a continuous buffer region around each anomaly, thus providing a more reliable and nuanced evaluation of model performance.

## D.5. Training Details

**Pre-training Setup**  The pre-training setup for our time series model is built around a transformer architecture. This encoder features 8 transformer layers, each with 8 attention heads. The internal dimension (`d_model`) of the model is 512, which is then projected down to a dimension of 256 (`d_proj`). The input time series is divided into patches of size 16. The model is trained using the `AdamW` optimizer with a learning rate of 5e-4 and a weight decay of 1e-5. The total loss is a sum of two components: a Mean Squared Error loss for a masked reconstruction task and a Cross-Entropy loss for an anomaly detection task. Training runs for a maximum of 50 epochs with a batch size that starts at 64 but is adjusted dynamically, and it includes an early stopping mechanism that halts training if the validation loss does not improve for 7 epochs.

**Pre-training Dataset Details**  We trained our model **only** on our custom synthetic dataset, which comprises a total of 2.5B data points. This dataset is composed of a vast number of individual time series with widely varying lengths. Figure 10 illustrates the distribution of these lengths. As the histogram shows, our pre-training curriculum follows a long-tailed distribution. It is dominated by a large volume of shorter series, which teaches the model common, localized patterns. Critically, it also includes a significant and diverse tail of longer series, with lengths extending up to 10,000 time steps. For multivariate series, our training data mixes channel counts of 1, 8, 12, 16, and 20 variables to ensure robustness across varying multivariate settings.

# E. Extended Results and Analysis

## E.1. Full Results

This section provides the complete, unabridged performance results that support the summary claims made in the main text.

**Experimental Setup and Zero-Shot Protocol**  Our evaluation involves two distinct comparisons: (1) a direct **zero-shot** comparison against other foundation models and lightweight statistical baselines, and (2) a comparison against **full-shot** models where specialized baselines are trained on the target data. For the main results in Table 1, TimeRCD uses a single fixed context window of 5000 for every dataset. Under this protocol, TimeRCD ranks first in **38** and second in **14** of the 64 zero-shot evaluation cases (16 datasets × 4 metrics). Against full-shot baselines, TimeRCD ranks first in **35** and second in **3** cases.

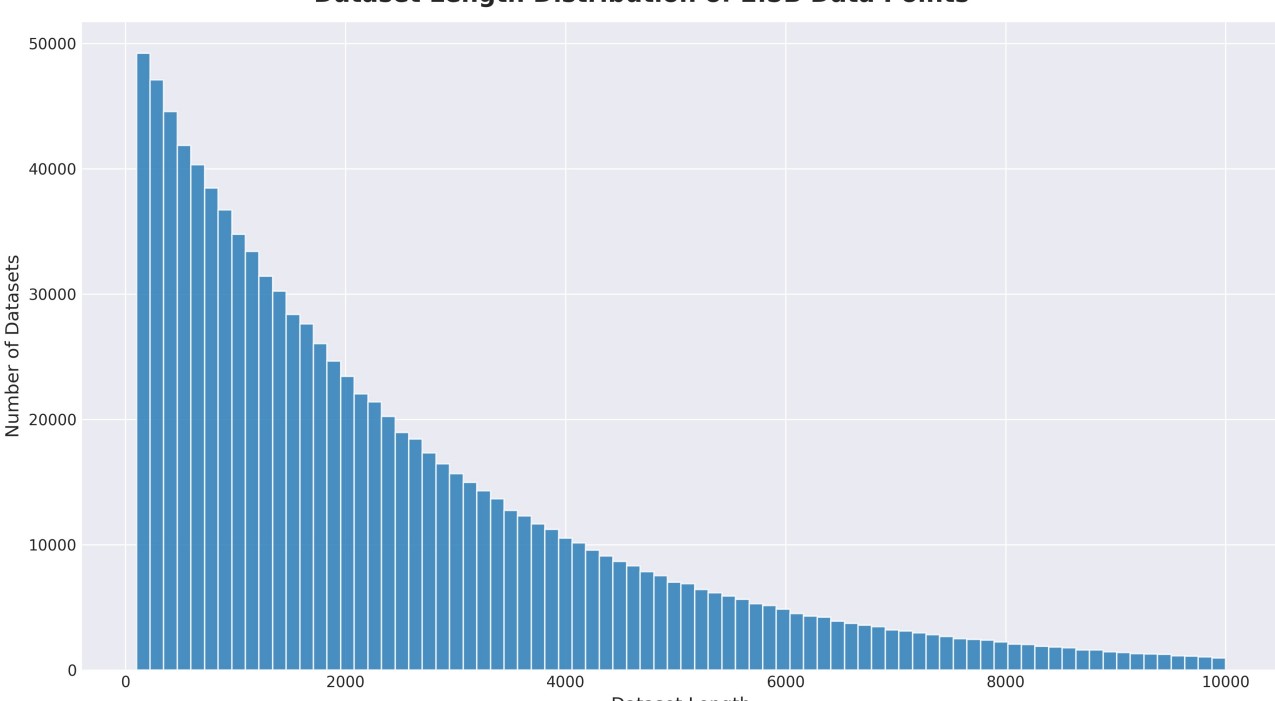

*Figure 10.* Distribution of time series lengths within our 2.5B point synthetic pre-training dataset. The histogram follows a long-tailed distribution, demonstrating that the pre-training curriculum is highly diverse. It contains a large volume of shorter series (under 4,000 steps) to teach common patterns, while also including a significant number of longer series (up to 10,000 steps) to ensure the model learns long-range temporal dependencies.

**Context Window Size Parameters for Main Results**    For the main results presented in Table 1, a fixed context window of **5k** is used for every dataset. Appendix E.3 reports window-size sensitivity only as an analysis of how performance changes with context length, not as the protocol used to select the main-table results.

### E.2. Contextual Analysis

To provide a controlled environment for testing our model's contextual understanding, we generated a new suite of specialized, unseen test datasets. The objective was to create two distinct sets: one containing only **point anomalies** and another containing only **contextual anomalies**. This section details their construction process.

**General Construction Process**    Each dataset was generated using the framework described in the main paper, with a crucial modification: only one type of anomaly was injected per set. We first synthesized a complex "normal" baseline series, combining trend, seasonal, and noise components. We then injected specific anomalies into these clean series to create our test cases. These datasets were not used in any part of the pre-training process.

**Point Anomaly Datasets**    Point anomalies are defined as short-term, localized deviations that can be identified by comparing a data point to its immediate neighbors, without requiring long-range context.

**Contextual Anomaly Datasets**    Contextual anomalies are defined as deviations from a long-term, established pattern, such as a disruption in a seasonal cycle. Detecting them requires the model to understand the global structure of the time series.

**Dataset Summary**    The final collection consists of two distinct sets of time series, with their key characteristics summarized in Table 10.

*Table 10.* Quantitative comparison on specialized point and contextual anomaly datasets. The table is grouped by anomaly type, with metrics listed vertically for each. TimeRCD is competitive on point anomalies and stronger on contextual anomalies on the reported metrics. Best results in each row are highlighted in **bold**.

| Type | Metric | TimeRCD | Chronos | DADA | MOMENT | TSPulse | Time MOE | TimesFM |
|---|---|---|---|---|---|---|---|---|
| **Point** | Affiliation-F | **0.912** | 0.899 | 0.823 | 0.886 | 0.733 | 0.793 | 0.889 |
| | $F1_T$ | **0.604** | 0.441 | 0.387 | 0.551 | 0.214 | 0.303 | 0.446 |
| | Standard-F1 | **0.521** | 0.352 | 0.121 | 0.472 | 0.090 | 0.149 | 0.397 |
| | VUS-PR | **0.666** | 0.293 | 0.396 | 0.538 | 0.063 | 0.190 | 0.385 |
| **Contextual** | Affiliation-F | **0.949** | 0.804 | 0.721 | 0.813 | 0.730 | 0.756 | 0.792 |
| | $F1_T$ | **0.781** | 0.174 | 0.227 | 0.284 | 0.210 | 0.269 | 0.154 |
| | Standard-F1 | **0.827** | 0.154 | 0.048 | 0.172 | 0.088 | 0.078 | 0.140 |
| | VUS-PR | **0.879** | 0.108 | 0.132 | 0.169 | 0.056 | 0.063 | 0.101 |

## E.3. Window Size Sensitivity Analysis

In this section, we show the detailed performance of our model across different context window sizes (from 1k to 13k) for each of the four primary evaluation metrics. The following figures provide a per-dataset breakdown, which supports the summarized analysis presented in the main paper.

The results in Figures 11 through 14 empirically validate our key finding: the optimal context length is task-dependent, and datasets with long-term or complex structures see a clear benefit from larger window sizes.

- **Positive Scaling Trend:** A positive association between window size and performance is evident for several datasets across most metrics. For example, **Power**, **UCR**, **MGAB**, **SMD**, and **SMAP** all demonstrate a general upward trend, especially on the F1-T, Standard-F1, and VUS-PR metrics. This suggests that for these series, a larger context provides a more robust understanding of "normality," leading to better anomaly detection.

- **Flat Performance on Short Series:** In contrast, datasets with inherently short time series, such as **YAHOO**, **NEK**, and **TODS**, show a flat performance profile. Their scores remain largely unchanged regardless of the window size. This is expected because the length of the time series itself acts as the natural context limit, and the implementation masks padded positions.

- **Task-Dependent Fluctuation:** Other datasets, like **NAB**, **PSM**, and **SWaT**, exhibit more fluctuating or non-monotonic behavior. For these, the ideal window size varies, with performance sometimes peaking at intermediate lengths. This underscores the value of having a flexible architecture where the context window can be tuned as a hyperparameter to achieve optimal results for a specific task.

- **Metric Sensitivity:** The Affiliation-F metric (Figure 11) generally appears less sensitive to changes in window size compared to the other three metrics. This suggests that while a larger context may be crucial for the precise, point-wise accuracy captured by metrics such as Standard-F1, the broader, proximity-based evaluation of Affiliation-F is less affected.

These detailed results reinforce the conclusion that the model's ability to handle extensive, variable-length contexts is an important feature for a general-purpose time-series anomaly detector.

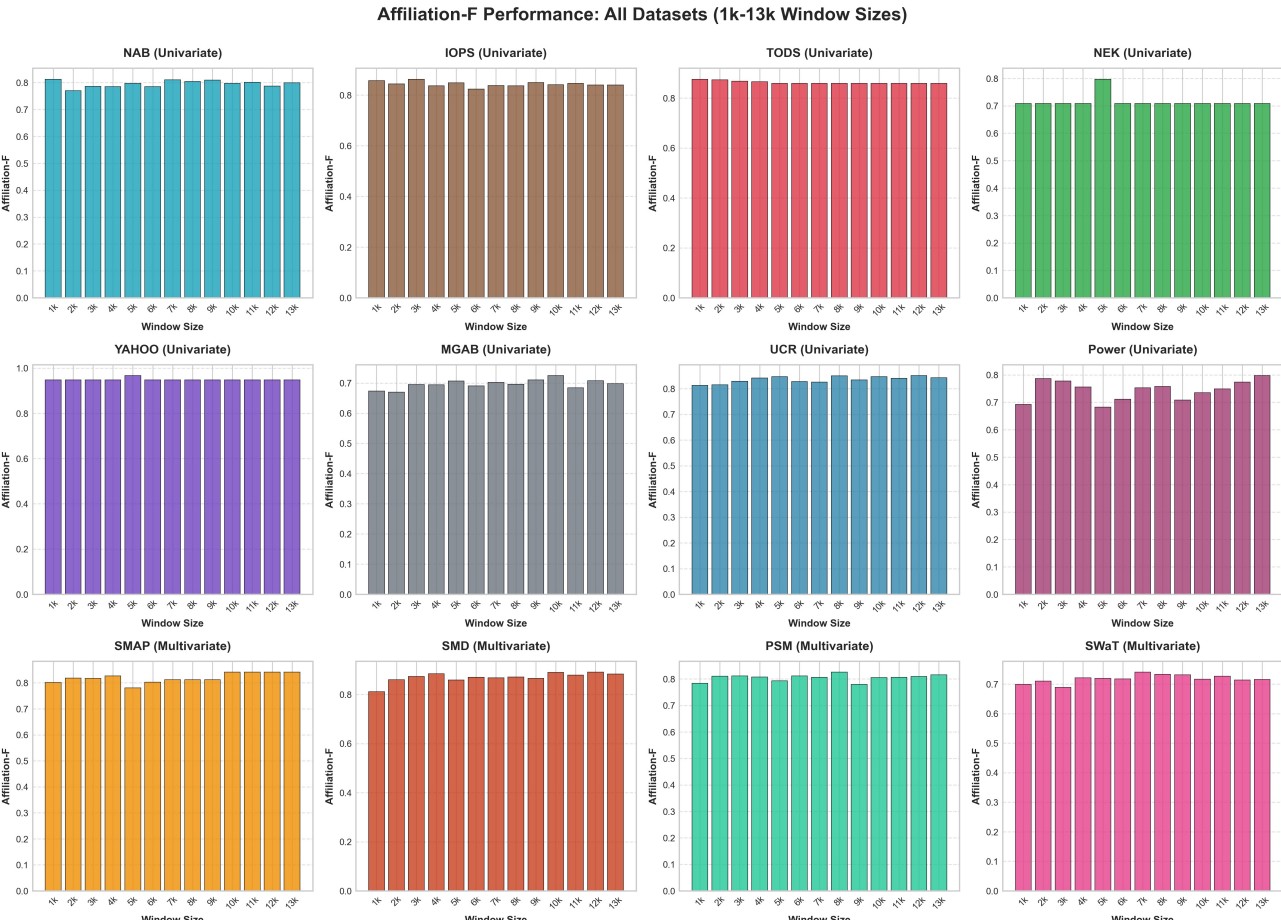

*Figure 11.* Affiliation-F performance across all 14 datasets as a function of the input window size from 1k to 13k.

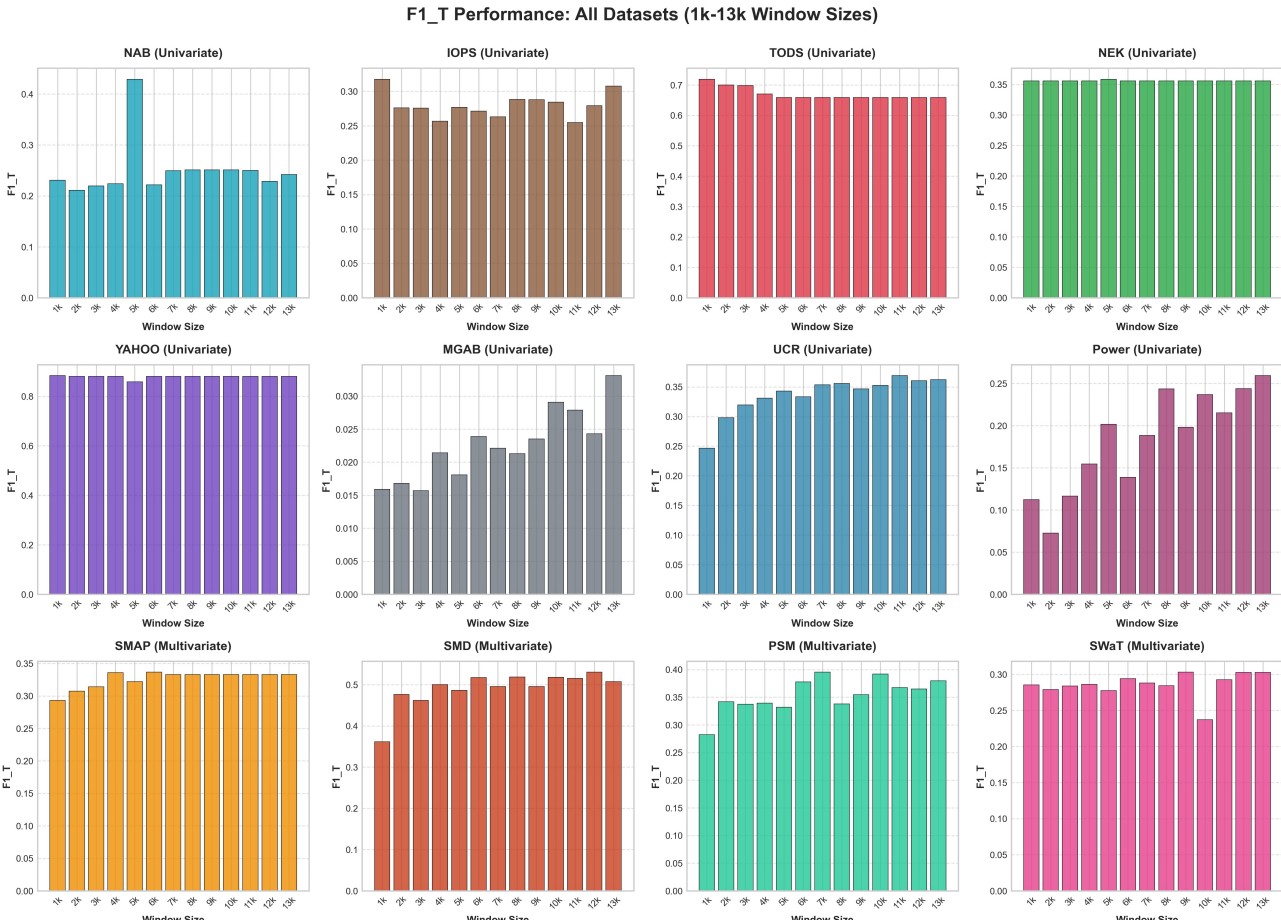

*Figure 12.* F1$_T$ performance across all 14 datasets as a function of the input window size from 1k to 13k.

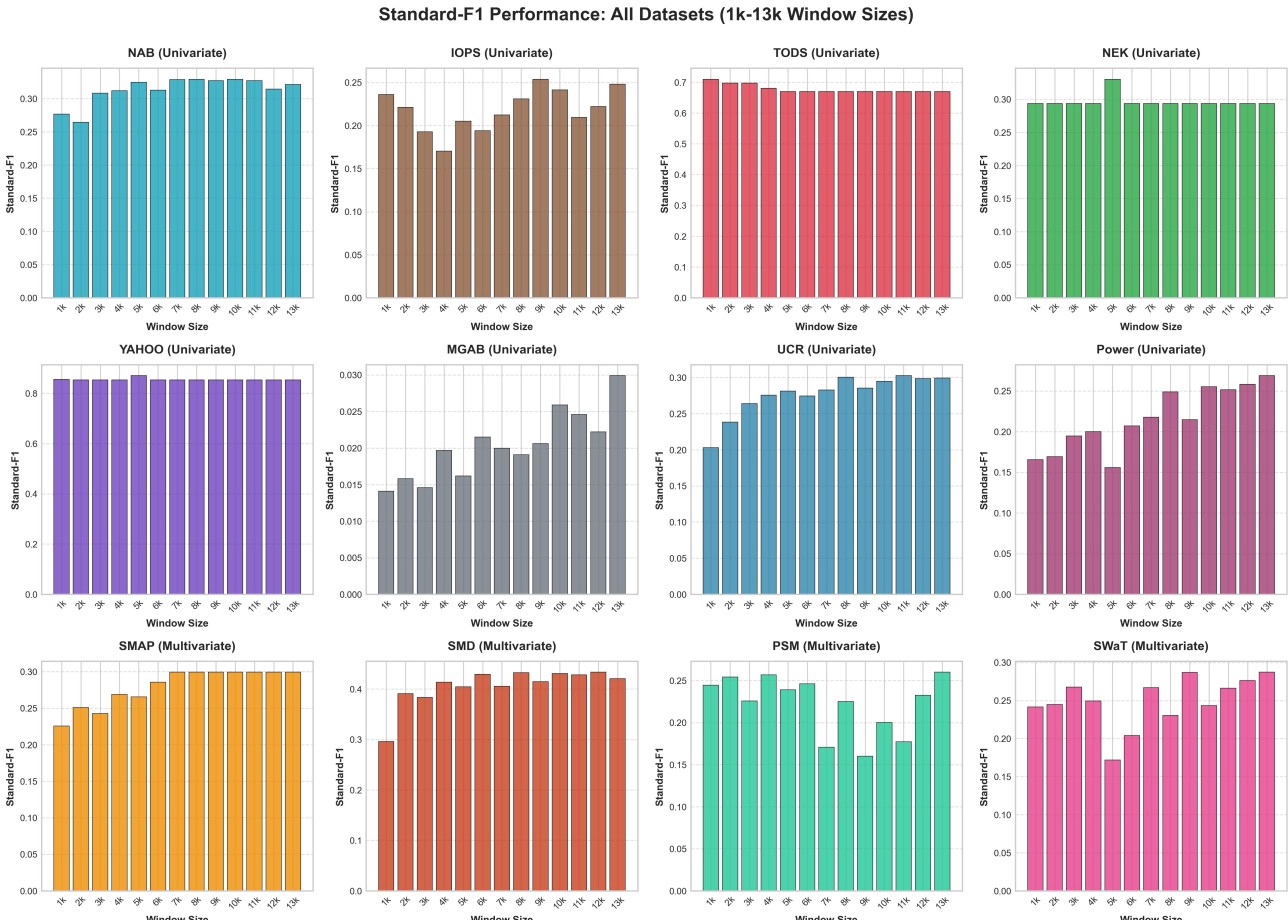

*Figure 13.* Standard-F1 performance across all 14 datasets as a function of the input window size from 1k to 13k.

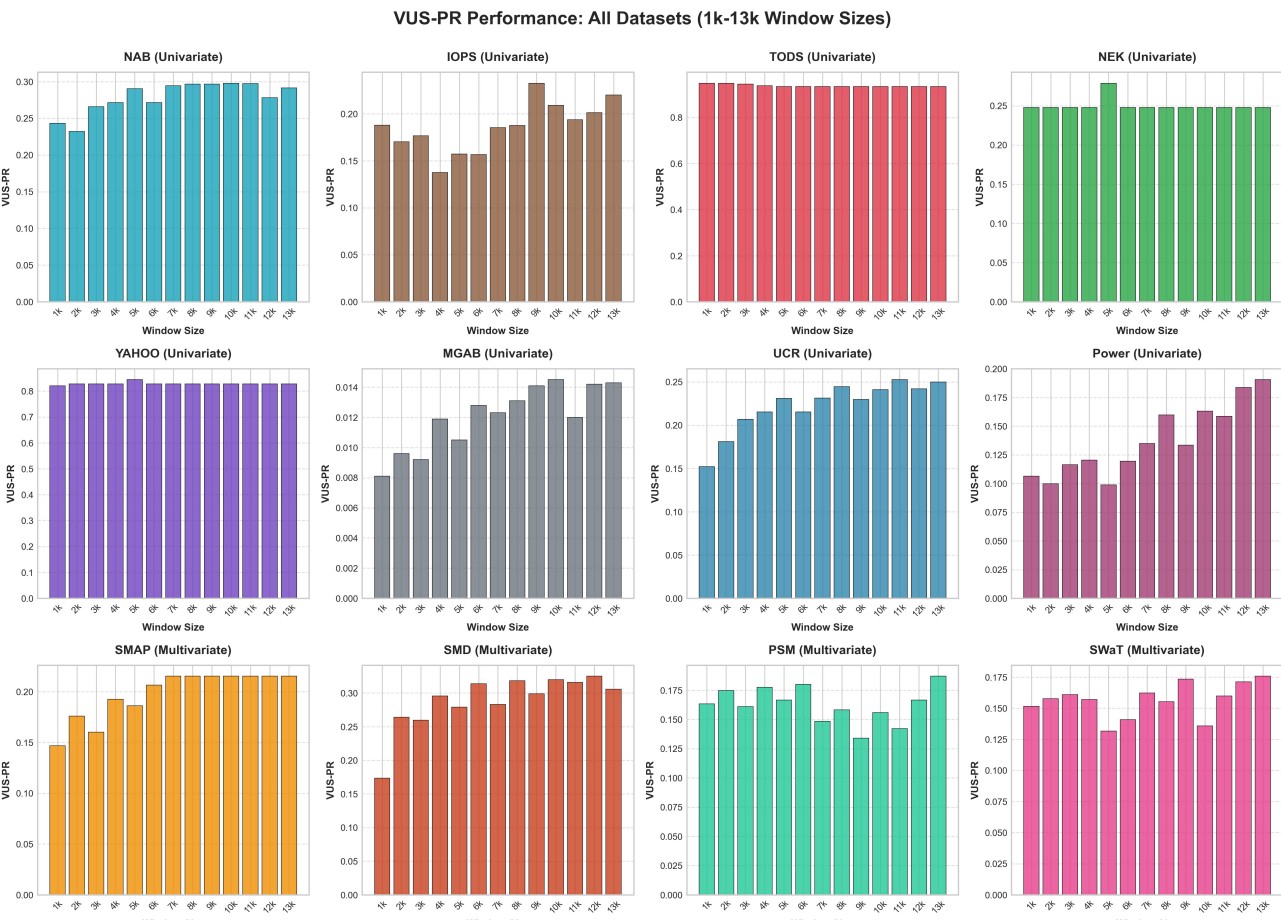

*Figure 14.* VUS-PR performance across all 14 datasets as a function of the input window size from 1k to 13k.

*Table 11.* Impact of Context Window Size: Performance comparison across available datasets. Best values are highlighted in **red**, second-best in blue.

| Metric | Model | Univariate Datasets | | | | | | | | | | | Total 1st | Total 2nd |
|---|---|---|---|---|---|---|---|---|---|---|---|---|---|---|
| | | IOPS | MGAB | NAB | NEK | Power | SED | TODS | UCR | YAHOO | Stock | WSD | | |
| **Affiliation-F** | **1k** | **91.34** | 67.15 | 89.60 | **83.95** | 66.26 | 81.19 | 85.03 | 80.55 | 94.30 | **48.81** | 91.93 | 03 | 01 |
| | **2k** | 89.18 | 68.38 | 90.31 | **83.95** | 75.63 | 90.47 | **85.72** | 81.81 | **95.94** | 46.25 | 91.81 | 03 | 01 |
| | **3k** | 89.87 | 72.80 | 92.71 | **83.95** | 83.05 | **95.12** | 85.43 | 83.47 | 95.94 | 43.45 | 92.06 | 03 | 02 |
| | **4k** | 89.92 | 73.34 | 93.39 | **83.95** | 80.87 | 90.00 | 84.91 | 83.54 | 95.94 | 42.96 | **92.20** | 03 | 00 |
| | **5k** | 86.98 | **74.54** | 93.81 | **83.95** | **85.02** | 88.99 | 84.86 | 84.34 | 95.94 | 41.31 | 91.00 | 03 | 01 |
| | **6k** | 89.70 | 74.07 | 93.53 | **83.95** | 79.68 | 92.22 | 84.86 | **86.43** | 95.94 | 41.37 | 89.45 | 03 | 02 |
| | **7k** | 88.93 | 71.36 | 93.85 | **83.95** | 81.67 | 88.46 | 84.86 | 85.62 | 95.94 | 41.62 | 89.28 | 02 | 00 |
| | **8k** | 88.26 | 73.32 | **94.11** | **83.95** | 82.14 | 79.72 | 84.86 | 85.88 | 95.94 | 41.61 | 89.90 | 03 | 00 |
| | **9k** | 89.74 | 73.68 | 94.06 | **83.95** | 75.22 | 82.28 | 84.86 | 85.67 | 95.94 | 41.21 | 89.39 | 02 | 01 |
| | **10k** | 89.36 | 72.03 | 93.87 | **83.95** | 79.66 | 80.53 | 84.86 | 85.23 | 95.94 | 40.53 | 89.72 | 02 | 00 |
| | **11k** | 90.11 | 73.69 | 93.94 | **83.95** | 83.70 | 82.62 | 84.86 | 85.52 | 95.94 | 41.19 | 90.38 | 02 | 00 |
| | **12k** | 89.48 | 70.78 | 93.82 | **83.95** | 82.83 | 80.52 | 84.86 | 86.03 | 95.94 | 41.52 | 89.16 | 02 | 00 |
| | **13k** | 89.62 | 71.37 | 93.81 | **83.95** | 82.61 | 82.88 | 84.86 | 86.39 | 95.94 | 42.11 | 90.19 | 02 | 01 |
| | **14k** | 90.27 | 69.92 | 93.96 | **83.95** | 81.84 | 82.75 | 84.86 | 85.70 | 95.94 | 42.30 | 90.27 | 02 | 01 |
| | **15k** | 88.38 | 72.47 | 93.96 | **83.95** | **87.18** | 78.77 | 84.86 | 86.17 | 95.94 | 42.70 | 89.99 | 03 | 00 |
| **F1_T** | **1k** | 44.51 | 1.82 | 40.41 | **76.91** | 21.19 | 42.45 | 67.87 | 30.64 | 76.39 | **23.28** | 41.35 | 02 | 01 |
| | **2k** | 40.81 | 1.74 | 42.66 | **76.91** | 19.84 | 56.63 | 69.22 | 33.72 | **79.14** | 21.21 | 43.16 | 02 | 02 |
| | **3k** | 44.15 | 4.07 | 48.41 | **76.91** | 19.84 | **66.26** | **69.35** | 38.40 | 79.14 | 19.59 | 45.04 | 04 | 01 |
| | **4k** | 46.62 | 7.18 | 49.26 | **76.91** | 25.18 | 54.52 | 66.55 | 38.67 | 79.14 | 19.08 | **45.07** | 03 | 01 |
| | **5k** | 45.39 | 9.05 | **50.49** | **76.91** | 23.05 | 60.59 | 68.46 | 41.18 | 79.14 | 18.38 | 44.13 | 03 | 01 |
| | **6k** | 43.02 | 10.33 | 49.24 | **76.91** | 22.09 | 54.29 | 68.46 | 43.01 | 79.14 | 18.72 | 39.58 | 02 | 00 |
| | **7k** | 41.58 | 11.34 | 50.18 | **76.91** | 24.80 | 49.63 | 68.46 | 42.97 | 79.14 | 18.25 | 37.37 | 02 | 00 |
| | **8k** | **46.92** | 12.40 | 50.08 | **76.91** | 27.30 | 41.71 | 68.46 | 43.38 | 79.14 | 18.53 | 36.93 | 03 | 00 |
| | **9k** | 44.80 | 11.03 | 50.11 | **76.91** | 22.26 | 46.14 | 68.46 | 44.07 | 79.14 | 18.64 | 36.41 | 02 | 00 |
| | **10k** | 42.98 | 12.06 | 50.13 | **76.91** | 29.84 | 45.47 | 68.46 | 43.04 | 79.14 | 18.52 | 36.85 | 02 | 00 |
| | **11k** | 42.49 | 15.40 | 50.29 | **76.91** | 32.16 | 49.70 | 68.46 | 43.91 | 79.14 | 18.76 | 37.73 | 02 | 01 |
| | **12k** | 44.49 | 12.54 | 49.88 | **76.91** | 33.95 | 46.84 | 68.46 | 43.85 | 79.14 | 18.67 | 37.45 | 02 | 00 |
| | **13k** | 43.48 | **16.73** | 49.63 | **76.91** | **35.90** | 51.35 | 68.46 | 44.35 | 79.14 | 19.21 | 38.69 | 03 | 02 |
| | **14k** | 44.33 | 15.91 | 50.09 | **76.91** | 33.86 | 51.66 | 68.46 | 43.90 | 79.14 | 19.37 | 38.23 | 02 | 01 |
| | **15k** | 45.40 | 14.98 | 50.12 | **76.91** | **36.53** | 49.60 | 68.46 | **44.89** | 79.14 | 19.63 | 38.15 | 04 | 00 |
| **Standard-F1** | **1k** | 40.31 | 1.71 | 30.78 | **68.64** | 20.70 | 42.68 | 65.47 | 24.64 | 86.19 | **24.67** | 43.14 | 02 | 01 |
| | **2k** | 39.33 | 1.50 | 33.04 | **68.64** | 17.82 | 56.88 | 68.47 | 27.29 | **89.05** | 22.54 | 44.14 | 02 | 02 |
| | **3k** | 42.70 | 3.72 | 40.22 | **68.64** | 15.44 | **66.59** | 69.39 | 31.87 | 89.05 | 20.83 | **45.73** | 04 | 01 |
| | **4k** | 48.66 | 6.62 | 41.73 | **68.64** | 25.23 | 54.78 | 66.71 | 31.72 | 89.05 | 20.24 | 43.88 | 02 | 00 |
| | **5k** | 47.28 | 8.63 | **42.66** | **68.64** | 23.12 | 60.85 | 70.02 | 33.47 | 89.05 | 19.56 | 41.04 | 04 | 01 |
| | **6k** | 46.43 | 9.28 | 41.49 | **68.64** | 21.32 | 54.70 | 70.02 | 36.09 | 89.05 | 19.77 | 36.57 | 03 | 01 |
| | **7k** | 45.76 | 10.27 | 42.54 | **68.64** | 24.54 | 49.98 | 70.02 | 35.02 | 89.05 | 19.29 | 35.79 | 03 | 00 |
| | **8k** | 47.97 | 10.36 | 42.36 | **68.64** | 27.33 | 42.02 | 70.02 | 35.71 | 89.05 | 19.59 | 36.09 | 03 | 00 |
| | **9k** | **49.68** | 9.45 | 42.48 | **68.64** | 21.90 | 46.53 | 70.02 | 35.98 | 89.05 | 19.64 | 36.13 | 04 | 00 |
| | **10k** | 48.24 | 11.94 | 42.50 | **68.64** | 29.90 | 45.73 | 70.02 | 34.98 | 89.05 | 19.47 | 35.86 | 03 | 00 |
| | **11k** | 47.20 | 14.95 | 42.57 | **68.64** | 32.24 | 50.08 | 70.02 | 35.58 | 89.05 | 19.75 | 36.15 | 03 | 00 |
| | **12k** | 47.95 | 10.81 | 42.27 | **68.64** | 34.02 | 47.13 | 70.02 | 35.85 | 89.05 | 19.65 | 35.65 | 03 | 00 |
| | **13k** | 47.75 | **17.00** | 41.94 | **68.64** | 36.03 | 51.77 | 70.02 | 35.99 | 89.05 | 20.25 | 36.80 | 04 | 01 |
| | **14k** | 48.99 | 14.00 | 42.50 | **68.64** | 33.73 | 51.86 | 70.02 | 35.87 | 89.05 | 20.37 | 36.41 | 03 | 01 |
| | **15k** | 48.75 | 16.64 | 42.60 | **68.64** | **36.45** | 49.90 | 70.02 | **36.91** | 89.05 | 20.65 | 36.77 | 05 | 02 |
| **VUS-PR** | **1k** | 35.82 | 0.70 | 28.73 | **71.13** | 11.29 | 37.63 | 92.61 | 19.40 | 83.32 | **76.67** | 32.64 | 02 | 02 |
| | **2k** | 32.00 | 0.91 | 32.52 | **71.13** | 10.22 | 63.47 | 92.19 | 20.34 | **87.88** | 74.92 | 35.99 | 02 | 01 |
| | **3k** | 38.48 | 1.86 | 41.23 | **71.13** | 11.07 | **82.47** | **93.44** | 25.91 | 87.88 | 73.79 | **37.49** | 05 | 00 |
| | **4k** | 41.43 | 2.72 | 42.15 | **71.13** | 14.24 | 60.52 | 90.57 | 24.61 | 87.88 | 73.19 | 36.72 | 02 | 01 |
| | **5k** | 42.98 | 3.09 | **44.65** | **71.13** | 14.63 | 69.86 | 91.67 | 27.48 | 87.88 | 72.73 | 29.95 | 03 | 01 |
| | **6k** | 39.31 | 3.45 | 42.33 | **71.13** | 13.67 | 67.65 | 91.67 | 30.03 | 87.88 | 72.81 | 26.25 | 02 | 00 |
| | **7k** | 39.85 | 3.72 | 44.43 | **71.13** | 16.63 | 58.93 | 91.67 | 29.15 | 87.88 | 72.71 | 24.99 | 02 | 00 |
| | **8k** | 41.76 | 3.90 | 44.26 | **71.13** | 17.25 | 45.83 | 91.67 | 30.17 | 87.88 | 72.61 | 25.40 | 02 | 00 |
| | **9k** | 43.13 | 3.47 | 44.44 | **71.13** | 13.89 | 52.28 | 91.67 | 30.39 | 87.88 | 72.93 | 25.06 | 02 | 01 |
| | **10k** | 42.58 | 3.96 | 44.36 | **71.13** | 19.51 | 51.41 | 91.67 | 30.05 | 87.88 | 72.58 | 25.52 | 02 | 00 |
| | **11k** | 41.96 | 4.60 | 44.37 | **71.13** | 21.39 | 56.05 | 91.67 | 30.32 | 87.88 | 72.85 | 25.84 | 02 | 00 |
| | **12k** | 42.32 | 3.62 | 43.85 | **71.13** | 22.19 | 50.37 | 91.67 | 30.71 | 87.88 | 72.57 | 25.94 | 02 | 00 |
| | **13k** | 42.54 | **5.40** | 43.70 | **71.13** | 24.29 | 57.42 | 91.67 | 30.80 | 87.88 | 73.25 | 26.01 | 03 | 02 |
| | **14k** | 43.77 | 4.20 | 44.16 | **71.13** | 22.47 | 56.01 | 91.67 | 30.76 | 87.88 | 73.18 | 26.49 | 02 | 01 |
| | **15k** | **43.93** | 4.88 | 44.20 | **71.13** | **25.67** | 55.60 | 91.67 | **31.35** | 87.88 | 73.07 | 27.02 | 05 | 01 |

## E.4. Additional Analysis for Data Scaling

In this section, we provide the full, per-dataset results for our investigation into the impact of pre-training data scale on model performance. These granular results support the weighted average trends and conclusions presented in the main paper.

**Experimental Setup**    To investigate the effect of pre-training data scale, we trained our model on increasingly larger subsets of our synthetic dataset: 350M, 700M, and the full 2.5B data points. Each model was trained using an identical protocol to isolate the effect of the data volume. Due to the predominantly univariate nature of the large-scale datasets used to construct our synthetic curriculum, this scaling analysis was conducted exclusively on the nine univariate benchmark datasets.

**Detailed Results and Analysis**    The results, presented in Table 12, show a positive empirical scaling trend. As the volume of pre-training data increases, the model's overall performance generally improves. This is most evident in the "Total 1st" place rankings, where the full **2.5B** model outperforms the smaller models across all four metrics, securing the top rank in 24 out of 36 total evaluations. While the general trend is clear, the per-dataset results reveal some nuances. The scaling is not perfectly monotonic for every dataset-metric combination. For instance, the **700M** model achieves the best F1-T score on the IOPS and YAHOO datasets, and the **350M** model shows strong performance on the NEK dataset. These variations likely stem from a combination of dataset-specific characteristics and the inherent stochasticity of the training process. Overall, the aggregate rankings support the value of a large-scale pre-training curriculum. They are also consistent with continued gains from additional pre-training data in this regime, although they do not establish a saturation point.

*Table 12.* Performance comparison across different models and metrics. Best results for each dataset-metric combination are highlighted in **red**, and second-best are in blue.

| Metric | Model | Datasets | | | | | | | | | Total 1st | Total 2nd |
|---|---|---|---|---|---|---|---|---|---|---|---|---|
| | | IOPS | MGAB | NAB | NEK | Power | SED | TODS | UCR | YAHOO | | |
| **Affiliation-F** | 2.5B | 84.79 | **70.69** | 79.70 | **79.73** | 68.30 | **96.87** | **85.91** | **84.63** | **96.65** | 06 | 01 |
| | 700M | 86.46 | 68.87 | **86.43** | 72.74 | **73.02** | 92.07 | 83.95 | 82.23 | 93.74 | 02 | 05 |
| | 350M | **88.24** | 68.19 | 86.21 | 79.27 | 63.30 | 89.21 | 83.22 | 82.49 | 92.35 | 01 | 03 |
| **F1_T** | 2.5B | 27.70 | **01.81** | 42.90 | 35.87 | **20.17** | 69.43 | 65.89 | **34.30** | 85.86 | 06 | 00 |
| | 700M | **40.56** | 01.49 | 27.82 | 42.31 | 05.13 | 59.02 | 64.07 | 28.88 | **86.88** | 02 | 05 |
| | 350M | 36.08 | 01.08 | 25.69 | **55.06** | 06.23 | 53.92 | 58.76 | 31.59 | 85.99 | 01 | 04 |
| **Standard-F1** | 2.5B | 20.51 | **01.62** | 32.46 | 33.05 | 15.59 | 69.88 | 67.01 | 28.13 | 87.02 | 06 | 01 |
| | 350M | **30.63** | 01.13 | 30.12 | **50.89** | **17.39** | 54.18 | 61.66 | 24.38 | 82.16 | 03 | 01 |
| | 700M | 30.08 | 01.51 | 32.17 | 42.68 | 12.63 | 59.31 | 66.45 | 22.71 | 83.92 | 00 | 07 |
| **VUS-PR** | 2.5B | 15.71 | **01.05** | 29.02 | 27.88 | **09.88** | 80.75 | 93.46 | 23.09 | 84.41 | 06 | 01 |
| | 700M | **26.51** | 0.7200 | **30.18** | 36.27 | 08.51 | 67.37 | 91.62 | 18.35 | 78.48 | 02 | 05 |
| | 350M | 24.07 | 0.7000 | 27.16 | **48.66** | 09.66 | 61.63 | 88.62 | 20.59 | 74.68 | 01 | 03 |

## E.5. Additional Analysis for Ablation Study

The controlled ablation on the anomaly head, reconstruction objective, and inference rule is reported in Table 3. This study uses the same backbone, tokenization, synthetic corpus, and comparable training budget, and therefore provides a cleaner comparison across training objectives and inference rules. In particular, it distinguishes reconstruction as a training-time scaffold from reconstruction error as an inference-time anomaly score.

# F. Statistical Analysis Summary

| Scenario | Affiliation-F | F1_T | Standard-F1 | VUS-PR | Avg Rank |
|---|---|---|---|---|---|
| **Zero-shot Univariate** | 2nd (MR=2.73, M=0.844) | 1st (MR=2.45, M=0.414) | 1st (MR=2.64, M=0.393) | 1st (MR=2.27, MD=0.243) | **1.8 / 7** |
| **Zero-shot Multivariate** | 1st (MR=1.2, M=0.829) | 1st (MR=1.2, M=0.397) | 1st (MR=1.2, M=0.322) | 1st (MR=1.4, M=0.233) | **1.2 / 7** |
| **Full-shot Univariate** | 1st (MR=1.82, M=0.844) | 1st (MR=2.27, MD=0.350) | 1st (MR=2.64, MD=0.320) | 1st (MR=2.55, MD=0.243) | **2.3 / 9** |
| **Full-shot Multivariate** | 4th (MR=3.2, MD=0.816) | 4th (MR=3.2, M=0.397) | 4th (MR=4.4, M=0.322) | 4th (MR=3.6, M=0.233) | **3.6 / 9** |

*Table 13.* TimeRCD Performance Summary across All Scenarios. Legend: MR = Mean Rank, M = Mean, MD = Median

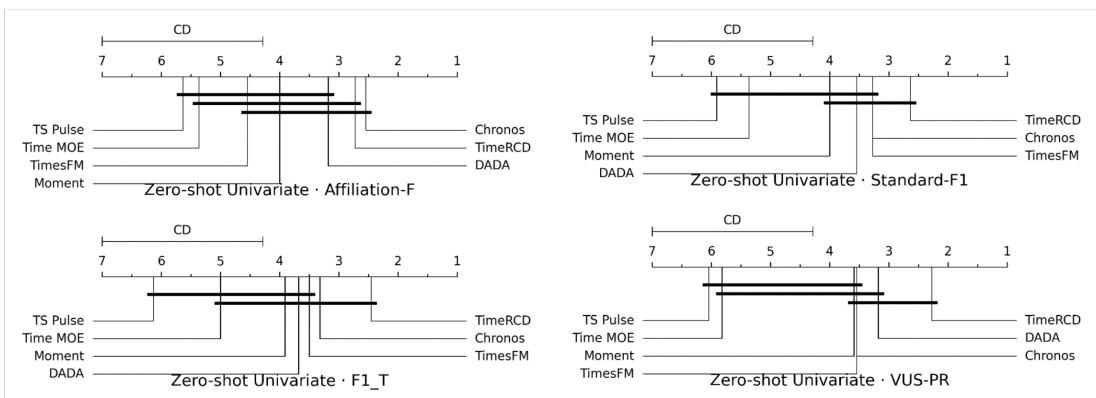

*Figure 15.* Statistical ranking visualization for zero-shot univariate anomaly detection.

# G. Efficiency and Scalability of TimeRCD

This appendix provides additional profiling results for TimeRCD in comparison with two baselines, TimeMOE and DADA. The goal is to characterize the computational feasibility, scalability, and memory behavior of TimeRCD across a wide range of sequence lengths, including regimes beyond those reported in the main text.

TimeRCD is evaluated on sequences up to 15,000 timesteps. For a sequence of 10,000 timesteps, TimeRCD achieves an inference latency of 21.8 ms, corresponding to a speedup of $12.6\times$ over TimeMOE and $95.2\times$ over DADA under the same hardware and profiling setup. Even at 15,000 timesteps, the inference time remains 86.22 ms, which is compatible with real-time or near real-time monitoring requirements in many practical applications. In our profiling setup, latency grows sublinearly over the measured sequence-length range.

In terms of memory, TimeRCD processes the entire input sequence rather than using a fixed-size sliding window. This design provides a longer effective receptive field and richer temporal context at the cost of increased memory usage with sequence length. At 10,000 timesteps, peak memory consumption is approximately 1.5 GB, which remains manageable on widely available GPUs. TimeRCD also uses $3.2\times$ fewer parameters than TimeMOE (35.5M vs. 113.4M), mitigating model storage and deployment overhead while maintaining favorable long-sequence efficiency.

## G.1. Inference Time (1k–10k Timesteps)

All models are profiled on a single GPU using isolated processes and a global warmup phase to reduce variability. Table 14 reports the end-to-end inference latency as a function of sequence length.

| Model | Seq=1K | Seq=2K | Seq=5K | Seq=10K |
|---|---|---|---|---|
| **TimeRCD** | **8.0** | **9.2** | **12.2** | **21.8** |
| TimeMOE | 36.5 | 66.7 | 130.8 | 274.2 |
| DADA | 207.6 | 423.1 | 1042.3 | 2074.0 |

*Table 14.* Inference time (ms) as a function of sequence length for TimeRCD and baselines.

Across all lengths, TimeRCD exhibits a much slower growth in latency compared to TimeMOE and DADA. This behavior is primarily attributed to its patch-based processing (patch size = 16), which effectively reduces the computational burden for long sequences while preserving temporal coverage.

## G.2. Peak Memory Usage and Parameter Counts

Table 15 summarizes peak memory usage for different sequence lengths together with the parameter counts of each model.

TimeMOE and DADA have approximately constant memory usage with respect to sequence length due to their use of fixed-size sliding windows, which limit the effective context. In contrast, TimeRCD processes the full sequence and therefore exhibits increasing memory consumption with longer inputs. Nevertheless, the resulting memory requirements remain well within the capabilities of common consumer-grade GPUs for the sequence lengths considered here.

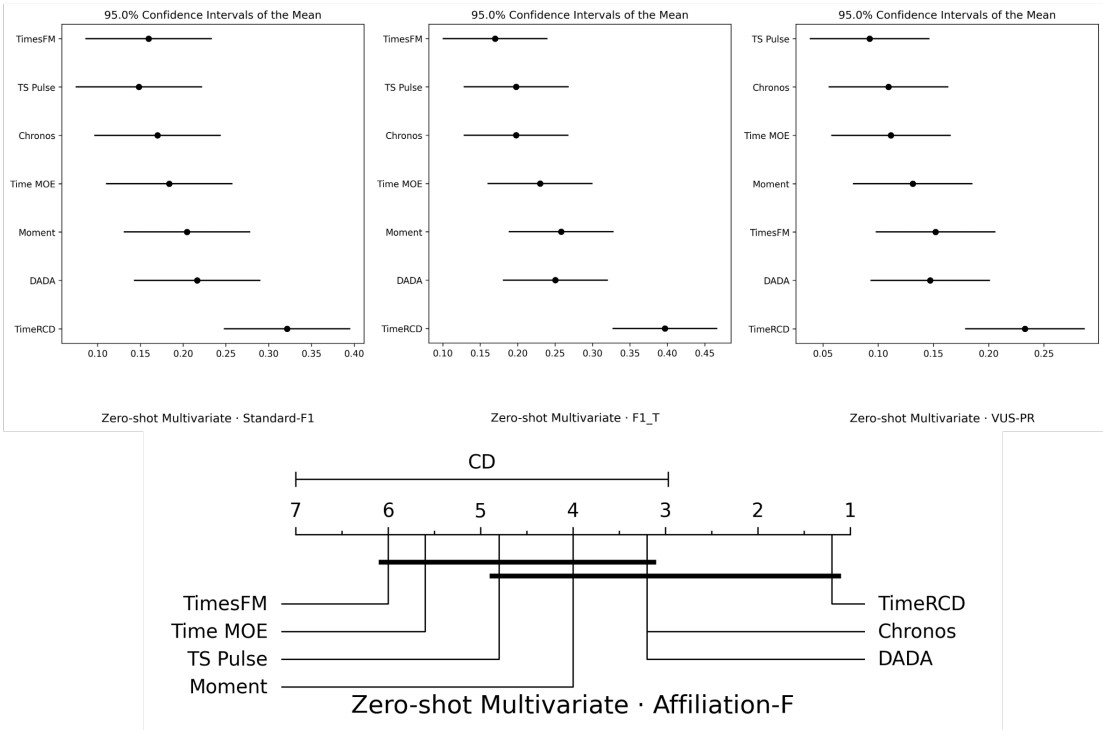

*Figure 16.* Statistical ranking visualization for zero-shot multivariate anomaly detection.

| Model | Seq=1K | Seq=2K | Seq=5K | Seq=10K | Parameters |
|---|---|---|---|---|---|
| **TimeRCD** | **295** | **434** | **858** | **1561** | **35.5M** |
| TimeMOE | 462 | 462 | 462 | 462 | 113.4M |
| DADA | 17.8 | 17.8 | 17.8 | 17.8 | 1.8M |

*Table 15.* Peak memory usage (MB) and parameter counts for TimeRCD and baselines.

## G.3. Scalability to Long Contexts (Up to 15k Timesteps)

To further characterize long-context behavior, TimeRCD is evaluated on sequences up to 15,000 timesteps. Table 16 reports the end-to-end inference latency, peak memory usage, and a normalized efficiency measure in milliseconds per 1,000 timesteps.

| Sequence Length | Inference (ms) | Memory (MB) | Efficiency (ms/1k steps) |
|---|---|---|---|
| 1,000 | 14.37 | 295 | 14.37 |
| 5,000 | 28.39 | 858 | 5.68 |
| 10,000 | 51.38 | 1561 | 5.14 |
| **15,000** | **86.22** | **2230** | **5.75** |

*Table 16.* Scalability of TimeRCD up to 15,000 timesteps.

The efficiency metric (milliseconds per 1,000 timesteps) decreases substantially when moving from 1,000 to 5,000 timesteps and then stabilizes around 5–6 ms per 1,000 timesteps for longer sequences. This pattern indicates that fixed overheads are amortized as the sequence length grows, leading to stable per-timestep cost in the long-context regime.

Overall, these extended results indicate that TimeRCD provides favorable latency and memory characteristics for long-context time series over the measured range, with a memory footprint suitable for both research and practical deployments.

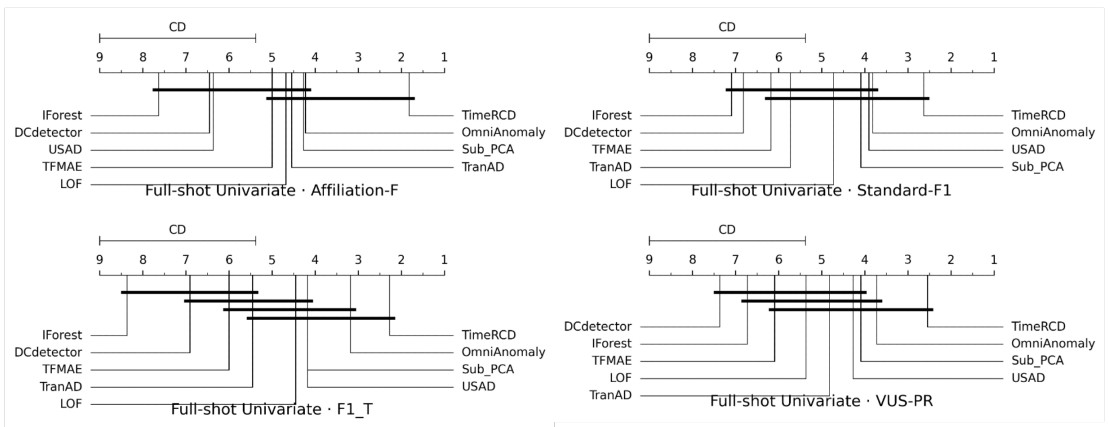

*Figure 17.* Statistical ranking visualization for full-shot univariate anomaly detection.

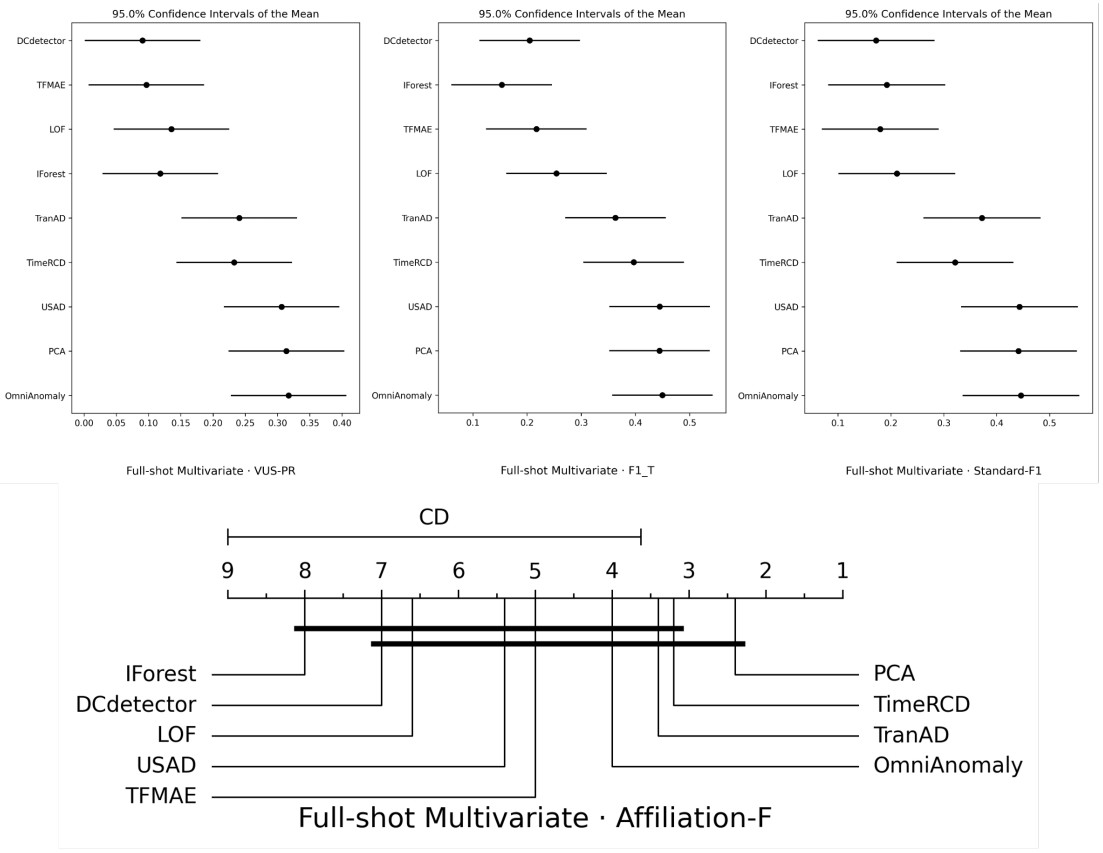

*Figure 18.* Statistical ranking visualization for full-shot multivariate anomaly detection.

