# OpenReview forum: "Towards Foundation Models for Zero-Shot Time Series Anomaly Detection: Leveraging Synthetic Data and Relative Context Discrepancy"
_ICML.cc/2026/Conference — ICML 2026 regular_

### Official Review · Reviewer_q7jU · 2026-03-04

**Soundness:** 3
**Presentation:** 3
**Significance:** 3
**Originality:** 4
**Overall Recommendation:** 5
**Confidence:** 3

**Summary:**

This paper introduces **TimeRCD**, a novel foundation model designed specifically for zero-shot Time Series Anomaly Detection. The paper argue that prevailing reconstruction-based TSAD models suffer from a fundamental "objective mismatch" and "representation conflict," as they tend to memorize static normal patterns from training data and consequently struggle when the definition of normality undergoes a concept shift across unseen domains. To overcome these limitations, the paper proposes a paradigm shift toward Relative Context Discrepancy (RCD), which explicitly trains a standard Transformer backbone to detect anomalies on-the-fly by contrasting a query token against its surrounding temporal context rather than relying on global priors. To empower this RCD objective without the biases of real-world data, the paper engineer a massive 2.5 B synthetic data. This Context-Dependent Data Generation process dynamically generates context-dependent anomalies to break the semantic coupling between local morphological shapes and anomaly labels. Extensive evaluations across 16 diverse public datasets demonstrate that TimeRCD achieves state-of-the-art zero-shot performance.

**Compliance With Llm Reviewing Policy:**

Affirmed.

**Final Justification:**

The authors provided detailed responses during the rebuttal process, and I have decided to maintain my original rating.

**Key Questions For Authors:**

To truly disentangle the contribution of the RCD model architecture from the sheer richness of the 2.5B synthetic corpus, the authors should train at least one representative reconstruction-based baseline **exclusively on the proposed synthetic data**.

**Limitations:**

As highlighted in the Weaknesses section, the most significant limitation is the confounding variable between the proposed RCD methodology and the newly introduced 2.5B synthetic dataset. Since the zero-shot baselines (e.g., MOMENT, TimesFM, DADA) are evaluated using models trained on their original, respective corpora, it is ambiguous to isolate how much of TimeRCD's superior performance is intrinsically due to the RCD objective versus the high-quality, causally-rich synthetic data.

**Strengths And Weaknesses:**

## Strengths

Originality & Significance:

The paper introduces a novel perspective on zero-shot TSAD by shifting the paradigm from global reconstruction to Relative Context Discrepancy. Furthermore, pioneering the use of a purely synthetic, large-scale curriculum for pre-training a TSAD foundation model is a significant departure from existing hybrid-data approaches. This decoupling of morphological features from normality provides fresh and valuable insights into resolving representation conflicts in cross-domain scenarios.

Soundness:

The empirical validation is solid and comprehensive. The authors rigorously benchmark TimeRCD across 16 diverse public datasets, comparing it against a wide array of state-of-the-art zero-shot foundation models and dataset-specific full-shot baselines. The inclusion of detailed ablation studies, particularly on dataset scaling laws and context window sensitivity, strongly reinforces the validity of the empirical claims.

Presentation:

The submission is clearly written, well-structured, the storytelling effectively motivates the fundamental objective mismatch in current reconstruction-based methods before logically transitioning into the formulation of the RCD solution.

## Weaknesses

Soundness:

While the zero-shot empirical results are impressive, there is a critical variable between the proposed RCD model architecture and the massive synthetic dataset. Without this controlled comparison, it remains ambiguous whether the performance gains stem fundamentally from the "capturing long-range contextual dependencies" or the diversity and causal complexity of the proposed pre-training data.

---

> ### Author Rebuttal · Authors · 2026-03-31
>
> **W1:** We agree that this ambiguity is important. Our claim is **not** that long-range contextual modeling alone is sufficient to explain the gains. Rather, our hypothesis is that the improvement comes from the combination of the **RCD-style formulation** and the **context-dependent synthetic supervision**: the synthetic corpus provides scalable supervision that breaks the coupling between local morphology and anomaly labels, while RCD trains the model to use context **relatively** rather than rely on reconstruction error or local patterns alone. We therefore view the paper's central contribution as this **combination**, while agreeing that the current evidence only partially disentangles the two ingredients.
> ***
> **Q1:**
> We thank the reviewer for this valuable suggestion. We agree that training a representative reconstruction-based baseline on our synthetic corpus would be the **cleanest controlled comparison**, because it would keep the pretraining data fixed while directly testing whether the gain comes from the **RCD-style detection formulation** rather than from the richness of the synthetic corpus alone. This exact experiment is still missing from the current submission, so we do not view the current evidence as fully resolving this question. That said, two existing results provide **partial but still informative evidence**: **Table 2** shows that the same model benefits from our synthetic corpus relative to other data sources, supporting the value of the data itself; **Table 12** compares TimeRCD against a reconstruction-based alternative under the same training data, partially isolating the detection formulation and remaining consistent with an advantage of RCD. We therefore view the current evidence as **suggestive rather than conclusive**, and agree that the reviewer’s proposed experiment would be the most direct way to test this disentanglement.

---

> > ### Author Rebuttal · Reviewer_q7jU · 2026-04-01
> >
> > I appreciate the authors' transparency in acknowledging that the fundamental disentanglement between the RCD architecture and the richness of the 2.5B synthetic dataset remains partially unresolved. I acknowledge that pre-training a foundation-level reconstruction-based baseline from scratch on the 2.5B dataset is computationally prohibitive within the restricted rebuttal window. Nevertheless, I maintain that the indirect evidence provided in the response remains insufficiently persuasive to resolve the core concern.
> >
> > Furthermore, upon inspecting the provided code, I observed that the pre-trained weights have been included. I would like the authors to clarify **whether they intend to release the source code for the context-dependent synthetic data generation engine or the 2.5B pre-training dataset itself. My final assessment and score will be further adjusted by the guaranteed level of reproducibility regarding the data generation process.**

---

> > > ### Author Response · Authors · 2026-04-01
> > >
> > > We thank the reviewer for this important follow-up. We agree that reproducibility is especially important here, given that the disentanglement between the **RCD-style formulation** and the richness of the synthetic corpus is not yet fully resolved by the current rebuttal evidence.
> > >
> > > During the rebuttal period, the most concrete step we can provide immediately is a **representative subset** of the synthetic pretraining data. Specifically, we are releasing an anonymous **500-sample subset** of the synthetic time series [here](https://anonymous.4open.science/r/TimeRCD-5BE1/dataset_500samples_Nonelen_1.0ratio_univariate_attr_set_batch_0001.pkl.zip), together with **visualizations** of first 100 samples [here](https://anonymous.4open.science/r/TimeRCD-5BE1/dataset_500samples_first_100_visualization.pdf), to make the data characteristics and generation outcomes more transparent. We view this as a rebuttal-period snapshot for inspection, rather than as a substitute for full reproducibility.
> > >
> > > More importantly, upon acceptance, we intend to **fully release** the synthetic-data resources used in this work, including:
> > >
> > > - the **generator code** for the context-dependent synthetic data pipeline,
> > > - the corresponding **generation configurations** and key settings, and
> > > - the **full synthetic pretraining dataset**.
> > >
> > > Our goal is to ensure that the community can reproduce the data generation process itself, inspect the resulting corpus, and more directly evaluate whether the observed gains arise from the synthetic supervision, the **RCD formulation**, or their combination.
> > >
> > >
> > > ***
> > >
> > > As an additional update addressing the reviewer's core concern about attribution, we conducted a set of controlled experiments on **UCR** using the same backbone, tokenization, synthetic corpus, and comparable training budget, while varying only the training objective and inference rule:
> > >
> > > | Experiment | Training Objective | Inference Rule | VUS-PR |
> > > | --- | --- | --- | ---: |
> > > | Exp1 | Reconstruction + Anomaly | Anomaly head | 33.57 |
> > > | Exp2 | Reconstruction + Anomaly | Reconstruction error | 4.08 |
> > > | Exp3 | Reconstruction only | Reconstruction error | 3.35 |
> > > | Exp4 | Anomaly only | Anomaly head | 5.90 |
> > > | Exp5 | Stage-2: reconstruction-only pretraining, frozen encoder, then anomaly-head training | Anomaly head | 26.11 |
> > >
> > > These additional results provide several useful observations within our current setup:
> > >
> > > - **Exp1 vs. Exp3.** Using the same synthetic corpus with a standard reconstruction-only objective does **not** recover the performance of the full method, which suggests that the gain is not explained by synthetic data scaling alone.
> > > - **Exp1 vs. Exp2.** Even under the same joint training, replacing the anomaly head with reconstruction error at inference causes a large drop (**33.57 -> 4.08**), supporting our point that reconstruction can be useful during training while reconstruction error is a poor inference-time anomaly score.
> > > - **Exp4 vs. Exp5.** Training the anomaly head alone from scratch yields only **5.90**, whereas initializing from a reconstruction-trained encoder and then training the anomaly head improves performance substantially to **26.11**. Together with the [training-loss curves](https://imgur.com/a/GVYjOPh), this suggests that reconstruction mainly provides a useful representation-learning and optimization scaffold for the anomaly objective.
> > > - **Exp3 vs. Exp5.** Starting from a reconstruction-trained encoder and then training an anomaly head improves performance from **3.35** to **26.11**, suggesting that once a useful representation has been learned, the anomaly head contributes an important task-specific anomaly decision rule beyond reconstruction error alone.
> > > - **Exp1 vs. Exp5.** Joint training still outperforms the two-stage frozen-encoder setting (**33.57 vs. 26.11**), suggesting that the remaining gain comes not only from adding the anomaly head, but also from jointly adapting the representation to the anomaly objective.
> > >
> > > We hope these controlled experiments provide more direct evidence, within the rebuttal window, that the observed gain is not attributable to synthetic data scale alone. We will incorporate this clarification and these new analyses into **Section 3** of the main paper. We thank the reviewer again for pushing us to strengthen both the attribution analysis and the reproducibility discussion.

---

### Official Review · Reviewer_5dCG · 2026-03-08

**Soundness:** 3
**Presentation:** 3
**Significance:** 3
**Originality:** 3
**Overall Recommendation:** 4
**Confidence:** 3

**Summary:**

This paper proposes TimeRCD, a zero-shot time-series anomaly detection model trained on a large synthetic corpus with an RCD objective. The main idea is to detect anomalies relative to the surrounding context rather than relying purely on reconstruction error. The paper reports competitive results and includes a lot of additional analyses.

**Compliance With Llm Reviewing Policy:**

Affirmed.

**Final Justification:**

My concern has been resolved, and I will maintain my positive score of 4.

**Key Questions For Authors:**

1. Can authors explain why some datasets perform significantly better under a certain context window setting?

2. Since performance depends on context length, what practical guidance can be given for choosing the context window?

**Limitations:**

yes

**Strengths And Weaknesses:**

Strength

1. The motivation is sensible. In zero-shot TSAD, whether a local pattern is anomalous often depends on context.

2. The experiment section is fairly broad, with benchmark comparisons, contextual-anomaly evaluations, scaling analysis, ablations, and efficiency results.

3. The availability of code is another positive point for reproducibility.

Weakness

1. The paper’s central framework is a zero-shot method. From the appendix, the main results appear to use dataset-specific context-window choices. Since context length seems to matter substantially for some datasets, this weakens the zero-shot claim.

2. Context window appears to be an important part of the method, yet the paper does not offer a clear zero-shot strategy for choosing it in practice.

2. The presentation quality is poorer than I initially thought. In particular, the paper still contains cleanup issues, such as 'are are' and 'andlinearly' in the  'Variate-Window Tokenization' paragraph on page 3.

---

> ### Author Rebuttal · Authors · 2026-03-31
>
> **W1:**
> We thank the reviewer for raising this important concern. We agree that, in the original submission, selecting the context window per dataset makes the zero-shot claim less direct. At the same time, we would like to clarify that this protocol was applied **uniformly to all methods**, including both TimeRCD and the baselines, so the comparison itself was fair. To address the reviewer’s concern more directly, we now report updated main results using a **single fixed context window of 5000 across all datasets**(new checkpoints are in https://anonymous.4open.science/r/TimeRCD-5BE1/). Under this unified setting, the updated model achieves the best **VUS-PR** on **12/16 datasets** among **zero-shot** methods and on **9/16 datasets** even when compared against **full-shot** methods (Fig. 1). Compared with the original submission, the updated model improves the average **VUS-PR** by **6.7 absolute points** on univariate benchmarks and by **4.0 absolute points** on multivariate benchmarks. Because this evidence comes from a retrained model with revised synthetic-data settings, we view it as **stronger supporting evidence** for the fixed-window zero-shot protocol rather than a perfectly isolated test of the context-window variable alone. Moreover, the new context-length ablation (Fig. 2) shows that, although performance varies across datasets, the **average performance remains competitive across a reasonably broad range of window sizes**, while some datasets are clearly more sensitive.
>
> Fig.1 (Main Table): https://imgur.com/a/d0TLe7g
>
> Fig.2 (Context-window ablation): https://imgur.com/a/pWY2CYe
> ***
> **W2:** We thank the reviewer for this important question. We agree that the current paper does not provide a sufficiently clear practical strategy for choosing the context window in a zero-shot setting. Based on our ablation in Fig. 2, a reasonable practical rule is that **smaller context windows are often preferable when anomalies are highly local or the series has weak periodic structure**, whereas **larger windows are more helpful when normality is defined by broader contextual patterns or stronger periodicity**. The **Stock** example in Fig. 3 further illustrates why this choice matters: with a very large window, a single context may contain many outliers, which can blur the local notion of normality and make detection harder. We therefore view a fixed default window as a practical zero-shot protocol, while acknowledging that some datasets are more sensitive and may benefit from task-specific adjustment.
>
> Fig.3: https://imgur.com/a/DnJW7r9
> ***
> **W3:**
> We thank the reviewer for pointing this out. We apologize for the presentation issues and typographical errors, including "are are" and "andlinearly". These are genuine cleanup errors rather than methodological ambiguities, and correcting them is straightforward.
> ***
> **Q1:**
> Can authors explain why some datasets perform significantly better under a certain context window setting?
>
> Yes. One factor is methodological: in our earlier synthetic data, we identified an artifact where anomalies were less likely to appear near the end of generated sequences. This could encourage the model to treat the tail of a context window as less likely to be anomalous, which in turn amplified sensitivity to context-window choice. We reduced this artifact in the updated synthetic-data generation used for Fig. 1. A second factor is data-dependent: some datasets contain highly local anomalies, while others require broader context to define normality. Consistent with this, Fig. 2 shows that although the best window can vary across datasets, the **average performance remains competitive across a reasonably broad range of window sizes**. We therefore view the fixed-window result as meaningful additional support for the zero-shot setting, while still acknowledging that context length remains an important practical factor.
> ***
> **Q2:**
> Since performance depends on context length, what practical guidance can be given for choosing the context window?
>
> Our practical recommendation is to use a **single fixed default window** as the zero-shot protocol, and we use **5000** in the updated results. When task-specific adjustment is possible, a useful rule of thumb is: **smaller windows** are often preferable when anomalies are highly local or the series has weak periodic structure (e.g., **Stock** in Fig. 2,3), whereas **larger windows** are more helpful when normality depends on broader contextual structure or stronger periodicity (e.g., **MGAB** and **Power**). In this sense, the fixed-window protocol is a practical default rather than a claim that context length is unimportant.

---

> > ### Author Rebuttal · Reviewer_5dCG · 2026-04-02
> >
> > Thank you for the detailed rebuttals. My concern about the context window has been resolved, and I will maintain my positive score.

---

> > > ### Author Response · Authors · 2026-04-05
> > >
> > > We sincerely thank the reviewer for the positive update. We are glad that our clarification on the context-window issue has satisfactorily resolved the concern. We also appreciate the reviewer’s constructive feedback, which helped us improve the paper.

---

### Official Review · Reviewer_w74D · 2026-03-10

**Soundness:** 3
**Presentation:** 3
**Significance:** 2
**Originality:** 2
**Overall Recommendation:** 4
**Confidence:** 4

**Summary:**

This paper proposes TimeRCD, a foundation model framework for zero-shot time series anomaly detection (TSAD). The authors argue that most existing time-series foundation models rely on reconstruction-based objectives, which create an objective mismatch: models are optimized to reconstruct typical patterns rather than explicitly distinguish anomalous behavior. To address this issue, the paper introduces Relative Context Discrepancy (RCD), a new training paradigm that detects anomalies by modeling relational differences between a query timestamp and its surrounding temporal context.

**Compliance With Llm Reviewing Policy:**

Affirmed.

**Final Justification:**

The author addressed my concerns during the rebuttal process, so I raised my score to 4. However, as I mentioned earlier, I hope the author can conduct a more in-depth analysis and consideration in the paper regarding the relationship and independent contributions between reconstruction and anomaly heads, the overall framework, and synthetic data. This will contribute to further improvement.

**Key Questions For Authors:**

See weaknesses.

**Limitations:**

Yes.

**Strengths And Weaknesses:**

Strengths:

- The central argument that reconstruction-based objectives may not align well with anomaly detection is well-motivated and supported by prior literature.

- The paper includes several empirical analyses (scaling laws, window-size sensitivity, anomaly-type evaluation) that provide insight into the behavior of the proposed approach.

Weaknesses:

- Leveraging attention interactions in Transformers to capture contextual irregularities is not a novel concept, while solely using synthetic data for pre-training will discard some of the characteristics present in the real-world datasets.

- The author claims that reconstruction is not a good objective in TSAD. However, TimeRCD still introduces reconstruction as an auxiliary training component in the training objective. Whether it is conducive to capturing temporal correlation or abnormality, it seems to indicate that reconstruction is still very important for the TSAD task. Table 12 in the appendix seems to confirm this point. The ablation study based solely on reconstruction actually performed better than the result based solely on the anomaly head.

- Some design choices such as reconstruction auxiliary loss appear somewhat heuristic and lack theoretical justification.

---

> ### Author Rebuttal · Authors · 2026-03-31
>
> **W1:** We thank the reviewer for this thoughtful comment. We agree that using attention to model contextual interactions is not, by itself, a novel idea, and this is not the main claim of our paper. Our contribution is instead the **RCD paradigm** for zero-shot TSAD: rather than relying on a fixed notion of global normality, we train a standard Transformer to detect anomalies through **relative contextual information**. To make this possible, we pair the model with a **context-dependent synthetic corpus** that explicitly breaks the coupling between local morphology and anomaly labels, allowing the model to learn when the same pattern should be treated as normal or anomalous under different contexts.
>
> Regarding synthetic-only pre-training, we agree that synthetic data cannot fully capture all characteristics of real-world datasets. However, prior work has highlighted that synthetic data are valuable precisely because they provide **scalable, diverse, and controllable supervision** when real data are limited or difficult to annotate[1,2,3]. This is especially relevant in zero-shot TSAD, where richly annotated real anomalies with diverse contextual roles are extremely scarce. Our goal is therefore not to mimic a specific real domain, but to provide scalable supervision for learning a more transferable principle of anomaly detection, namely context-relative abnormality.
>
> Moreover, synthetic pre-training is complementary to real-data adaptation. As **preliminary complementary evidence**, we have so far completed fine-tuning experiments on **four representative datasets**, and all four show improved **VUS-PR**. Here, the **Before Finetuning** scores correspond to the updated zero-shot results in [**Fig. 1**](https://imgur.com/a/d0TLe7g), obtained with the revised synthetic data generation settings:
>
> | Dataset | Before Finetuning | After Finetuning | Improvement |
> | --- | ---: | ---: | ---: |
> | NAB | 44.65 | 50.32 | +5.67 |
> | NEK | 71.13 | 75.51 | +4.38 |
> | Stocks | 72.73 | 80.19 | +7.46 |
> | Power | 14.63 | 20.02 | +5.39 |
>
> Fig.1: https://imgur.com/a/d0TLe7g
>
> These preliminary results suggest that synthetic pre-training can provide a useful initialization, while real data can further adapt the model to domain-specific characteristics.
>
> [1] Comprehensive Exploration of Synthetic Data Generation: A Survey
>
> [2] Empowering Time Series Analysis with Synthetic Data: A Survey and Outlook in the Era of Foundation Models
>
> [3] CauKer: Classification Time Series Foundation Models Can Be Pretrained on Synthetic Data
> ***
> **W2:** We thank the reviewer for this insightful comment. We agree that our current wording in the Introduction and Methodology was not sufficiently precise. Our criticism is not of reconstruction itself as a training objective. In fact, masked reconstruction is a useful way to learn general temporal representations. What we question is the practice of using reconstruction error directly as the anomaly score at inference time, which is common in many reconstruction-based anomaly detection methods, especially in unsupervised settings. A key distinction of TimeRCD is that the anomaly decision is made by a supervised anomaly head, while masked reconstruction is used only as an auxiliary objective for representation learning during training.
>
> Regarding Table 12, we did try training with only the anomaly detection head. However, this setting is much harder to optimize. A likely reason is that the BCE supervision is relatively sparse, which makes it difficult for the model to learn meaningful RCD-related representations from the anomaly objective alone. In this sense, the masking reconstruction loss serves as an auxiliary training signal that stabilizes representation learning. This does not contradict our main point: reconstruction can be helpful during training, while reconstruction error may still be suboptimal as the inference-time anomaly score.
> ***
> **W3:** We thank the reviewer for this important point. The role of the masking reconstruction auxiliary loss deserves clearer discussion. In our view, this design is motivated by widely used self-supervised learning practice: masked reconstruction provides a strong training signal for learning informative temporal representations and can be understood as a denoising-style objective that encourages the model to capture underlying temporal structure. This is also consistent with several time-series works that use masked objectives to learn transferable representations[4-6]. In our framework, its primary role is to improve optimization and representation learning, rather than to define the anomaly score itself. We agree that a more rigorous theoretical understanding of its exact contribution remains an important limitation of the current work.
>
> [4] MOMENT: A Family of Open Time-series Foundation Models
>
> [5] TimeMAE: Self-Supervised Representations of Time Series with Decoupled Masked Autoencoders
>
> [6] Ti-MAE: Self-Supervised Masked Time Series Autoencoders

---

> > ### Author Rebuttal · Reviewer_w74D · 2026-04-02
> >
> > I appreciate the authors' detailed responses, which have partially addressed my concerns. The clarification on the distinction between reconstruction as a training objective versus reconstruction error as an inference-time anomaly score (W2) is helpful and makes the paper's position more nuanced. However, several points remain insufficiently resolved.
> >
> > The preliminary fine-tuning results on four datasets are encouraging, but they also raise a concern. The improvements after fine-tuning are quite significant, which suggests that the zero-shot representations learned from synthetic data alone may still have a notable gap compared to what is achievable with real data. Could the authors comment on how many real-world labeled samples are used during fine-tuning to achieve these gains?
> >
> > The authors acknowledge that training with only the anomaly detection head is "much harder to optimize" and attribute this to sparse BCE supervision. This is a reasonable hypothesis, but it also suggests that the anomaly head alone may not be learning sufficiently discriminative RCD-related features without the reconstruction signal. This raises a deeper question: to what extent is the anomaly head genuinely capturing contextual irregularities versus relying on features primarily shaped by the reconstruction objective? Have the authors conducted any analysis to disentangle the contributions of the two objectives to the final anomaly detection performance?
> >
> > The most important concern remains regarding the attribution of the reported performance improvements. It is currently unclear whether the gains stem from the proposed RCD framework and its specific objectives or if they are primarily a consequence of the large-scale synthetic dataset. To isolate the contribution of the methodology, it is necessary to provide a controlled baseline comparison. Specifically, training a standard foundation model on the same synthetic dataset would clarify whether the "RCD paradigm" provides a significant delta over simple data scaling.

---

> > > ### Author Response · Authors · 2026-04-05
> > >
> > > We thank the reviewer for these thoughtful follow-up questions. Regarding the fine-tuning results, we agree that the protocol should be stated more clearly. In these preliminary adaptation experiments, we used the **TSB-AD validation set** for fine-tuning, which corresponds to about **20%** of the data under the benchmark split, and evaluated on the corresponding **evaluation set**. We therefore view these results as evidence that synthetic pretraining provides a useful initialization that can be further adapted with downstream labeled data, rather than as a few-shot result. We agree that the gap between zero-shot and fine-tuned performance is important, and we will clarify this protocol explicitly to avoid overstating what these experiments show.
> > >
> > > We also agree that the role of the anomaly head versus the reconstruction objective deserves a more careful analysis. To examine this, we ran additional controlled experiments on **UCR** and also inspected the [training-loss curves](https://imgur.com/a/GVYjOPh). The loss curves show that reconstruction loss converges stably in both the joint model and the reconstruction-only model, while the BCE/anomaly loss decreases steadily in the joint setting but remains substantially higher and noisier when training the anomaly head alone from scratch. This supports the reviewer's concern that sparse anomaly supervision alone is difficult to optimize in our current setup. At the same time, when we first train a reconstruction-only encoder, freeze it, and then train only the anomaly head (**Exp5**), the anomaly optimization becomes much more stable and the UCR **VUS-PR** recovers substantially to **26.11**, compared with only **5.90** for anomaly-only training from scratch (**Exp4**). We therefore do not claim that the anomaly head alone is sufficient from scratch; rather, the current evidence suggests that reconstruction plays an important role as a representation-learning and optimization scaffold, after which the anomaly head can extract substantially more useful anomaly decisions.
> > >
> > > Most importantly, to better address attribution, we performed a controlled same-backbone / same-tokenization / same-synthetic-corpus comparison on **UCR**. All of these additional rebuttal experiments use the same revised synthetic-data setting as our updated main results; therefore, their absolute values may differ somewhat from those in the original submission, but the key point here is the controlled relative comparison within a unified setting:
> > >
> > > | Experiment | Training Objective | Inference Rule | VUS-PR |
> > > | --- | --- | --- | ---: |
> > > | Exp1 | Reconstruction + Anomaly | Anomaly head | 33.57 |
> > > | Exp2 (**Same Training, not use RCD**)| Reconstruction + Anomaly | Reconstruction error | 4.08 |
> > > | Exp3 (**reconstruction based TSFM**)| Reconstruction only | Reconstruction error | 3.35 |
> > > | Exp4 | Anomaly only | Anomaly head | 5.90 |
> > > | Exp5 | Stage-2: reconstruction-only pretraining, frozen encoder, then anomaly-head training | Anomaly head | 26.11 |
> > >
> > > - **Exp1 vs. Exp3.** Using the same synthetic corpus with a standard reconstruction-only objective does **not** recover the performance of the full method, suggesting that the gain is not explained by data scaling alone.
> > > - **Exp1 vs. Exp2.** Even under the same joint training, replacing the anomaly head with reconstruction error at inference causes a large drop (**33.57 -> 4.08**), which supports our claim that reconstruction can be helpful during training while reconstruction error is a poor inference-time anomaly score.
> > > - **Exp4 vs. Exp5.** Training the anomaly head alone from scratch yields only **5.90**, whereas initializing from a reconstruction-trained encoder and then training the anomaly head improves performance substantially to **26.11**. This suggests that reconstruction primarily provides a useful representation-learning and optimization scaffold for the anomaly objective.
> > > - **Exp3 vs. Exp5.** Starting from a reconstruction-trained encoder and then training an anomaly head substantially improves performance (**3.35 -> 26.11**), suggesting that once a useful representation has been learned, the anomaly head provides an important task-specific anomaly decision rule beyond reconstruction error alone.
> > > - **Exp1 vs. Exp5.** Joint training still outperforms the two-stage frozen-encoder setting (**33.57 vs. 26.11**), suggesting that the remaining gain comes not only from adding the anomaly head, but also from jointly adapting the representation to the anomaly objective.
> > >
> > > We will present these additional analyses with appropriate care and incorporate this discussion into **Section 3** of the main paper. We thank the reviewer again for the helpful suggestions.

---

### Official Review · Reviewer_DdyL · 2026-03-11

**Soundness:** 2
**Presentation:** 3
**Significance:** 2
**Originality:** 3
**Overall Recommendation:** 4
**Confidence:** 3

**Summary:**

The paper proposes a foundation model for zero-shot time-series anomaly detection that replaces reconstruction-based inference with a supervised anomaly classification head trained on synthetic data. The work's motivation is guided by observations of how reconstruction-based objectives suffer from objective mismatches and representation conflicts. TimeRCD instead learns to flag anomalies by comparing each window against its surrounding context, using a standard transformer trained on a synthetic corpus of over 2 billion data points generated through a multi-stage pipeline covering multiple different anomaly types. Experiments show strong zero-shot performance, particularly on contextual anomalies where competing methods largely collapse.

**Compliance With Llm Reviewing Policy:**

Affirmed.

**Final Justification:**

I increased my score to 4 during the rebuttal period to reflect the additional experiments and detailed clarifications provided by authors.  As I mentioned in the comments, I only hope the authors will take the time to improve the writing wherever questions or misconceptions arose in this round of reviewing.

**Key Questions For Authors:**

- Do the main results hold if a single fixed context window size were used across all datasets?
- How does TimeRCD compare against simple baselines such as moving-window variance?
- Have the authors considered forecasting-based anomaly detection as an additional baseline category?

**Limitations:**

The discussion of limitations is fairly minimal. More insight into failure modes and practical implications of certain design choices would strengthen the paper.

**Strengths And Weaknesses:**

**Strengths:**

- The paper's core argument of moving beyond reconstruction-based objectives is compelling, and the results validate the core claim that reconstruction misses long-range context.

- The synthetic data generation pipeline is well-designed and a meaningful contribution to the wider community. The distinction between exogenous and endogenous anomaly injection, combined with the causal DAG coupling across variates, makes the approach reproducible and extensible for future work.

- The evaluation covers both zero-shot and full-shot settings across multiple datasets, and I also appreciated the ablation studies that help tease apart the contributions of the individual components.

**Weaknesses:**

- For the zero-shot evaluation, it appears that the context window size was selected per dataset for the main results - does this end up being some form of hyperparameter tuning on the test benchmarks? The sensitivity analysis in the appendix is appreciated, but I think this point deserves more prominent discussion in the main paper.

- Unless I missed them, the paper does not compare against simple statistical baselines such as moving-window variance. Recent work (e.g., "When Foundation Models are One-Liners", ICLR 2026) shows that existing foundation models do not significantly outperform such trivial baselines on TSAD, and the five foundation models evaluated there overlap directly with TimeRCD's baselines. Given that TimeRCD leans more heavily on classification rather than reconstruction, it would be good to affirm improvements over these baselines. Some of the anomalies in Figure 5 also appear relatively straightforward to detect, which makes this comparison feel more relevant.

- The authors draw a clear distinction between reconstruction as a pretext task versus an inference metric, and the ablation confirms that reconstruction during training is essential for the anomaly head to function. This is not contradictory to the paper's argument, but I think it nuances the positioning against reconstruction-based methods in a way that deserves to be better described.

- I found the limitations discussion to be quite thin. The conclusion is a single paragraph, and the only limitation mentioned is the standard transformer backbone. More insight into failure modes, the dependency on reconstruction during training, and the per-dataset window tuning would be valuable, particularly for practitioners looking to build on this work.

- If I understood correctly, some of the synthetic contextual test datasets were generated using the same framework used for pre-training. The authors state that the specific instances were not used during training, but the generation process itself is shared, which may still give TimeRCD an advantage over real-world datasets?


**Score updated to 4 post-rebuttal.**

---

> ### Author Rebuttal · Authors · 2026-03-31
>
> **W1 & Q1:** We thank the reviewer for this important question. In the original submission, the context window was selected from several candidate values on each benchmark for **both TimeRCD and the baselines**, so this was indeed a form of benchmark-level tuning rather than a fully fixed zero-shot setting. To address this concern more directly, we updated the synthetic data generation process to reduce artifacts in the training data, and report updated main results using a **single fixed context window of 5000 across all datasets**(new checkpoints are in https://anonymous.4open.science/r/TimeRCD-5BE1/). The updated model achieves the best **VUS-PR** on **12/16 datasets** among **zero-shot** methods (Fig. 1).
>
> In addition, the new context-length ablation in Fig. 2 shows that, although the best window may vary across datasets, the **average performance remains competitive across a reasonably broad range of window sizes**. We therefore view the fixed-window result as meaningful additional support for the zero-shot setting, while still acknowledging that context length remains an important practical factor.
>
> Fig.1: https://imgur.com/a/d0TLe7g
>
> Fig.2: https://imgur.com/a/pWY2CYe
> ***
> **W2 & Q2:** We thank the reviewer for this helpful suggestion. We have added a simple statistical baseline based on **moving-window variance**, and the new results in Fig. 1 show that **TimeRCD outperforms it in VUS-PR by a clear margin**.
> ***
> **W3:** We thank the reviewer for this insightful comment. We agree that our current presentation does not make this distinction sufficiently clear. Our criticism is **not** of reconstruction itself as a training objective: masked reconstruction remains a useful way to learn general temporal representations. What we question is the practice of **using reconstruction error directly as the anomaly score at inference time**, which is common in many reconstruction-based TSAD methods. In contrast, in TimeRCD the primary anomaly decision is made by the anomaly head, while reconstruction is used only as an auxiliary signal for representation learning during training.
> ***
> **W4:** We thank the reviewer for this helpful suggestion. Below we summarize the main limitations that are most relevant to interpreting the current results.
>
> (1) **Failure modes.** The **Stock** dataset (Fig. 3) illustrates a representative failure case where TimeRCD performs less well. A likely reason is that, under a large context window, the window can contain around 200 anomalous points, so the model may partially treat these frequent outliers as part of the local reference pattern. This failure mode is consistent with the context-relative nature of our method: when anomalies become too frequent within the same window, the implicit notion of normality can become less reliable.
>
> Fig.3: https://imgur.com/a/DnJW7r9
>
> (2) **Per-dataset window tuning.** Although TimeRCD is reasonably robust to the context window length, the best choice can still vary across datasets. In particular, for datasets with more localized anomalies, smaller context windows can help the model focus more precisely on local deviations (e.g., Stock dataset in Fig. 2,3). In contrast, for datasets with stronger contextual structure, larger windows are often more helpful because they provide richer contextual information (e.g., MGAB and Power datasets).
>
> (3) **Role of reconstruction.** As discussed in our response above, reconstruction is used in TimeRCD only as an auxiliary training signal for learning useful temporal representations. The primary anomaly decision is made by the anomaly head rather than by reconstruction error at inference time.
> ***
> **W5:** We thank the reviewer for this important question. We agree that pre-training on synthetic data may transfer general knowledge of recurring anomaly patterns, as structures such as spikes and outliers also appear in real-world data. Our goal is to use synthetic data to provide scalable supervision for such patterns, while acknowledging that synthetic-to-real concerns remain. This issue is not unique to our method: approaches such as DADA may also benefit from similar shared structure, yet TimeRCD still outperforms them. This suggests that the improvement comes also from RCD and supervised learning on synthetic data.
> ***
>
> **Q3:** We thank the reviewer for this helpful suggestion. We have already included **TimesFM** and **Chronos**, which are representative forecasting-based TSFMs, in our comparisons. We further evaluated TimeRCD against methods reported in **TSB-AD**(https://thedatumorg.github.io/TSB-AD/), which includes several forecasting-based TSAD baselines, where the results in **Fig. 4** shows that TimeRCD is competitive with these methods.
>
> Fig.4: https://imgur.com/a/r9ZCIDq

---

> > ### Author Rebuttal · Reviewer_DdyL · 2026-04-01
> >
> > Thank you for the detailed rebuttals across all reviews, especially for running the additional evaluation using fixed context windows and comparing against simpler baselines. The clarification to W3 is also reasonable so I would just encourage emphasising this more.
> >
> > Thank you for also listing some of the failure modes associated with the model - again, I suggest including these clearly in the paper not only in the interest of transparency but to also help guide where future work is needed. I think the paper would still benefit from more discussion of its open questions, particularly around disentangling RCD from synthetic data quality. The fact that other methods share a similar setup doesn't fully address this, and I think it's an important direction for future work.
> >
> > In light of the above, I have increased my score to a 4.

---

> > > ### Author Response · Authors · 2026-04-01
> > >
> > > Thank you very much for the careful follow-up and for taking the time to revisit our rebuttal. We sincerely appreciate your positive reassessment and are encouraged that the added fixed-window evaluation, the comparison to simpler baselines, and the clarification of W3 helped address your main concerns.
> > >
> > > We also appreciate your constructive suggestions on what should be emphasized more clearly in the paper. In the revision, we will make the failure modes more explicit, strengthen the discussion around the role of reconstruction versus inference-time scoring, and more clearly discuss the remaining limitations around disentangling the contribution of RCD from synthetic-data design.
> > >
> > > Thank you again for the thoughtful feedback and for the score update.

---

### Decision · Program_Chairs · 2026-04-30

**Decision:**

Accept (regular)

**Comment:**

The paper introduces TimeRCD, a foundation model for time-series anomaly detection built on Relative Context Discrepancy (RCD), in which the model learns to identify anomalies by contrasting a query with its surrounding context, rather than by reconstructing inputs against fixed priors. A large, diverse synthetic corpus with context-dependent anomaly labels is constructed. Experiments show that TimeRCD outperforms both general-purpose and anomaly-specific foundation models in zero-shot TSAD.

The paper identifies a limitation of reconstruction-based TSAD foundation models and proposes a well-motivated alternative. The RCD paradigm reframes anomaly detection as relational contrast against surrounding context, which more directly matches the operational definition of anomaly in many real-world time series. Coupling this with a synthetic pre-training corpus enables effective supervised pre-training where labeled real data is scarce. The contribution is well positioned.